# Assessing Gender Bias in Particle Physics and Social Science Recommendations for Academic Jobs

**Robert H. Bernstein** [1,2,*]🔾, **Michael W. Macy** [3]🔾, **Wendy M. Williams** [2]🔾, **Christopher J. Cameron** [3]🔾, **Sterling Chance Williams-Ceci** [4]🔾 and **Stephen J. Ceci** [2]🔾

1   Particle Physics Division, Fermi National Accelerator Laboratory, P.O. Box 500, Batavia, IL 60510, USA
2   Department of Psychology, Martha Van Rensselaer Hall, Cornell University, Ithaca, NY 14853, USA;
    mwmacy@cornell.edu (M.W.M.); sjc9@cornell.edu (S.J.C.)
3   Department of Sociology, Uris Hall, Cornell University, Ithaca, NY 14853, USA;
    wendywilliams@cornell.edu (W.M.W.); cjc73@cornell.edu (C.J.C.)
4   Department of Information Science, Gates Hall, Cornell University, Ithaca, NY 14853, USA;
    scw222@cornell.edu
*   Correspondence: rhbob@fnal.gov

**Abstract:** We investigated gender bias in letters of recommendation as a possible cause of the under-representation of women in Experimental Particle Physics (EPP), where about 15% of faculty are female—well below the 60% level in psychology and sociology. We analyzed 2206 letters in EPP and these two social sciences using standard lexical measures as well as two new measures: author status and an open-ended search for gendered language. In contrast to former studies, women were not depicted as more communal, less agentic, or less standout. Lexical measures revealed few gender differences in either discipline. The open-ended analysis revealed disparities favoring women in social science and men in EPP. However, female EPP candidates were characterized as "brilliant" in nearly three times as many letters as were men.

**Keywords:** underrepresentation of women; gender bias in science; letters of recommendation

## 1. Introduction

Are women in physics limited by less enthusiastic depictions of their ability relative to their male counterparts? Does gender bias influence how job letters are written, resulting in women physicists having a more difficult time entering the academic workforce? The underrepresentation of women in math-intensive fields such as physics, engineering, computer science, economics, and mathematics is a problem that is historically persistent and extensively studied (Hill et al. 2010; National Academy of Science et al. 2007; National Research Council 2010; Valian 1998; Xie and Shauman 2003). Possible causes include hiring and promotion biases (Eaton et al. 2020; Moss-Racusin et al. 2012; Sheltzer and Smith 2014), leaky-pipeline issues (Adamo 2013; Goulden et al. 2011; Skibba 2019), differences in career preferences (Kelchtermans and Veugelers 2013; Su and Rounds 2015; Wang et al. 2013), and differential persistence/retention (Ceci et al. 2014; Kaminski and Geisler 2012; Martinez et al. 2017). Recent studies have examined possible gender bias in letters of recommendation, noting that "there is little research that addresses whether letters of recommendation for academia are written differently for men and women and whether potential differences influence selection decisions in academia" (Dutt et al. 2016; Li et al. 2017; Madera et al. 2009; Messner and Shimahara 2008; Schmader et al. 2007; Trix and Psenka 2003). Specifically, there is no research on gender differences in recommendations that compares academic fields in which women are well-represented to fields in which they are not.

This study investigates gender bias in letters of recommendation by comparing two fields differing dramatically in women's representation: experimental particle physics (EPP) and developmental social science. Women are well-represented among PhDs in the

developmental social sciences, which in this study is comprised of psychology and sociology applicants. In 2019, the overall representation of women in PhDs in developmental psychology was 83.1% and 63.8% of PhDs in sociology. In contrast, women remain significantly underrepresented in EPP (only 13.4% of PhDs and 15–16% of PhDs and faculty). For more details on the demographic breakdowns of our samples, see Appendix A, along with National Science Foundation (2019) and Porter and Ivie (2019).

We chose EPP (often called High-Energy Physics, or HEP) among many possibilities in the natural sciences since a large sample of letters was readily available over a multi-year timespan. We did not extend it beyond that discipline to reduce systematic errors that could arise from mixing EPP with other sub-fields of physics or other natural sciences. We had a similar large-statistics, convenient set of developmental psychology and sociology letters (about 80% psychology) and again chose not to extend the fields of study beyond that sample. We use "gender" to differentiate males from females except in a few instances for two reasons: (1) prior literature uses "gender bias" and "gender differences" and (2) if there is bias on the basis of sex, we assume that it arises from expectations of social roles, in which case "gender" is the more appropriate term.

### 1.1. Theory and Motivation for Methods

The study was guided by two goals: (1) could we reproduce prior findings and address their limitations and (2) could we find new methods and ask new questions that would provide useful insights? A review of existing literature can be found in Appendix B.

#### 1.1.1. Techniques in Prior Research

Previous research has (a) counted the number of words in a letter and (b) used word lists in pre-determined categories and counted their rates of usage. One study found "… female applicants are only half as likely to receive excellent letters versus good letters compared to male applicants" (Dutt et al. 2016). There has also been extensive investigation into "standout" and "grindstone" adjectives. A summary for physicists explains that standout adjectives are words such as "outstanding," "amazing," and "unmatched" and concludes (Blue et al. 2018):

> Standout words, which portray a candidate as talented and exciting, are most often found in letters of recommendation for men. Grindstone words, which create the impression that a candidate works hard but is not intellectually exceptional, are more often used for women.

This interpretation of "hard-working" as a backhanded compliment is supported in an influential paper that examined standard lexical measures (Trix and Psenka 2003):

> Of the letters for female applicants, 34 percent included grindstone adjectives, whereas 23 percent of the letters for male applicants included them. There is an insidious gender schema that associates effort with women, and ability with men in professional areas. According to this schema, women are hard-working because they must compensate for lack of ability.

Another line of research examined "agentic" and "communal" terms. Prior research found that agentic terms were more frequent in letters for men and communal terms were more frequent in letters for women. Agentic behaviors at work include speaking assertively, influencing others, and initiating tasks. Corresponding communal behaviors include being concerned with the welfare of others (i.e., descriptions of kindness, sympathy, sensitivity, and nurturance), helping others, accepting others' direction, and maintaining relationships (Madera et al. 2009).

Unfortunately, the above studies were limited by small samples, few measures, and the inability to completely rule out confounds from applicant characteristics as an alternative explanation for gender differences in the strength of letters. Our study addresses these limitations in three ways:

1. We compared letters written for candidates in two disciplines differing dramatically in the underrepresentation of women. We hypothesize that letters in EPP, a male-dominated field, will show more bias than ones with gender parity, psychology and sociology in the developmental social sciences. Prior studies have not made such a cross-discipline comparison.

2. It is of course possible that EPP female candidates are stronger than male candidates and male letter-writers downgrade their letters. We have investigated a sub-sample of letters with writers for a given candidate from both genders; any such difference would be apparent in this sample. We therefore tested for differences between female and male writers using a restricted sample of 918 letters that were written on behalf of 234 candidates with letters from both male and female non-primary advisors (excluding the PhD committee chair). This method allowed direct comparisons of differences in how women and men recommenders depicted the same candidates, thereby ruling out gender differences in candidate accomplishments as an explanation for differences in letters written by female versus male recommenders. We found no greater gender differences between writers for the restricted sample than for the entire sample of 2206 letters, and we therefore report results for the entire sample. With this additional comparison, not previously made in the literature, we can then attribute any observed disciplinary difference to gender bias.

3. We used the same measures of letter content derived from the Linguistic Inquiry and Word Count (LIWC) dictionaries used in many previous studies: the proportion of total words in the letter that appear in the lists of "achievement" words connoting accomplishment and success, words associated with "ability," and words that convey affective attraction and aversion called "posemo" and "negemo" in LIWC, or "positive affect" and "negative affect" in the literature (Pennebaker et al. 2015). However, this study used larger samples with a wider array of measures than those in any previous individual paper, including the four measures in previous research described above: "agentic," "communal,", "standout," and "grindstone" (Madera et al. 2009; Schmader et al. 2007; Trix and Psenka 2003). This allowed a simultaneous comparison of all these categories on a single sample, removing systematic errors from comparing different categories in different sets of letters belonging to different disciplines written across multiple decades – our study uses letters written in a single period from 2011–2017.

1.1.2. New Questions and Techniques

This study asks what appear to be four new questions:

1. Do men and women prefer to ask writers of the same sex for their letters? This question is different from a lexical analysis and focuses on the behavior of the candidate rather than the words chosen by the recommender. If such "gender homophily" exists, then its source is of interest. Furthermore, gender homophily could be a confounding effect in the lexical analyses: if male and female writers write letters differently, and men and women prefer letters from their own gender, then we may incorrectly attribute differences in the lexical analysis results to bias rather than gender differences in writing style.

2. Consider some word or word category that is a potential source of gender bias. We can measure one of two different quantities. We can determine the number of times one of our word categories appears in a letter divided by the number of words in the letter; we will call that rate-per-word, and that is what is normally reported. We can also measure the number of times a word category appears in a letter and count the fraction of letters containing that word category, whether the word category appears once or multiple times. What is the difference between a measurement of the rate per word and a measurement of the fraction of letters? We know some words such as "brilliant" have a high impact on the reader and a single use can have a large effect. Other words, such as those describing teaching, may send a message by accumulation—if you write about someone as a teacher more than as a researcher, no



single word has a large effect but the accumulation of such words can send a message. "Research" is mentioned one or more times in nearly every letter, and hence there is a strong signal in writing a letter that never mentions "research". Despite the effect sizes in many studies being about one word per letter, it has been argued that "only one statement can make a difference" (Madera et al. 2009). In contrast, if a writer uses "research" twice in one letter, that does not necessarily indicate that the writer was more enthusiastic compared to using that same term only once but we cannot determine that from this study.

3. Is there a bias in the rank of the writer? The rank of the letter-writer can also send a signal; letters from prestigious sources might carry more weight than other sources. We therefore examined the status of the author.

4. Are there limitations in using pre-existing word lists? Every discipline has its own signaling phrases, and lists for individual studies chose words and classifications targeted for their discipline and questions. This has led to a given word being assigned to different categories in different papers: several "grindstone" words are classified differently in different papers (Madera et al. 2009; Schmader et al. 2007). A general list such as LIWC 2015 chose and classified words in a manner that might not capture effects that require targeted classification. It would be best to let the data "speak for itself". We therefore performed an "open-ended" analysis where the actual words used in our sample were examined to see if they were associated with male or female candidates.

## 2. Materials and Methods

### 2.1. Data

We collected 2206 recommendation letters written on behalf of candidates for positions at the assistant professor level over multiple job searches between 2011 and 2017. These letters were written for EPP positions at Fermi National Accelerator Laboratory (963 letters for 206 men; 198 letters for 39 women) and for developmental social science positions in psychology and sociology at Cornell University (440 letters for 163 men; 605 letters for 222 women); the breakdown of letters by gender of writer and gender of candidate is given in Tables 1 and 2). For typical advertisements and other background information about EPP, see the webpage for Fermi National Accelerator Laboratory (Fermi National Accelerator Laboratory 2020).

All letters were stripped of salutations, addresses, complimentary closings, university branding, and similar text, reducing letter length by about 100 words. Letterhead "branding" often contained agentic or standout words, which would confound our measurements if not removed. Removing this content was relatively simple for the EPP letters. All EPP letters were in PDF in a standard format and were altered using Adobe Acrobat often by simply removing text boxes. Social Science letters came in different formats and the above processes were carried out by-hand.

**Table 1.** Descriptive Statistics for sample sizes used in this study. About 10% of EPP candidates submitted multiple applications over the period of the study. In those cases we only used the last application.

| Candidate | Male Writer | | Female Writer | | Sum | Total Male Candidates | Total Female Candidates |
| | Male | Female | Male | Female | | | |
|---|---|---|---|---|---|---|---|
| EPP | 842 | 176 | 121 | 22 | 1161 | 206 | 39 |
| Social Science | 301 | 298 | 139 | 307 | 1045 | 163 | 222 |

**Table 2.** Demographics of writers and candidate choice. Uncertainties are $1\sigma$ statistical (R.M.S./$\sqrt{N}$).

|  | EPP | Social Science |
|---|---|---|
| Letters/Male Candidate | 4.67 | 2.70 |
| Letters/Female Candidate | 5.08 | 2.73 |
|  |  |  |
| Female Writers/Male Candidate | 0.59 | 1.34 |
| Female Writers/Female Candidate | 0.56 | 1.38 |
|  |  |  |
| Male Writers/Male Candidate | 4.09 | 1.85 |
| Female Writers/Female Candidate | 4.51 | 1.34 |
|  |  |  |
| Female Writers/Male Writers | | |
| Male candidates | $0.14 \pm 0.014$ | $0.46 \pm 0.04$ |
| Female Candidates | $0.13 \pm 0.028$ | $1.03 \pm 0.084$ |
| Expected from Pool of Writers | $0.16 \pm 0.022$ | $0.67 \pm 0.04$ |

Applicants in either discipline are not required to give birthdates but both sets of CV's contained the candidate's work history, including their date of PhD. The typical EPP candidate was finishing their second three-year post-doctoral appointment after receiving their PhD. In the social science sample there was an occasional gap between baccalaureate and PhD, e.g., when a candidate worked a few years before starting graduate school (this almost never occurred in EPP.) The modal social science candidate had no postdoctoral fellowships. Candidates in both disciplines were about 25–30 years old.

*2.2. Existing Word List Method*

We used two sources for the word lists in this study: (a) three categories—Posemo (positive emotion/affect words), Negemo (negative emotion/affect words), and achievement—were obtained from the LIWC 2015 word-list and were not changed; (b) five additional lists (ability, agentic, communal, standout, and grindstone) were taken from three published sources, to provide comparability with past studies of letter content (Madera et al. 2009; Pennebaker et al. 2015; Schmader et al. 2007). We also added domain-specific words in EPP: for example, it is common to say that a candidate is in "the top 5%" as a standout phrase, and a few such phrases were added to the lists. Prior studies did not use domain knowledge to add such powerful signifiers.

The lists used are available in the Supplemental Materials. Our letters as modified above were then processed by LIWC2015.

*2.3. Open-Ended Analysis Methods*

Our open-ended analysis is not dependent on the words in word-lists but is based instead on any word with gendered associations. Between 30% and 86% of the patterns in each word-list category matched a word in the letter text but over 75% of the words in the base vocabulary were not found in any word-list. Therefore, because a great majority of the words in the letters are not in the word-list categories, it is possible for large differences in the letters to be overlooked using the lexical measures in previous studies.

The screening process began by algorithmically removing "stop words," personal pronouns, and words with little or no difference in gendered usage, leaving a set of 405 words that were potential sources of gender bias. Two of the senior authors, the physicist and one of the sociologist/psychologists, then independently (without knowing each other's choices) eliminated words that did not signal recommendation, without knowing the gender distribution. This included elimination of terms for which a gender difference might indicate gender-specific subfield specializations (e.g., "family" and "neuroscience" in developmental social science letters for women and men, respectively). It also included elimination of words with overtly context-dependent meaning (e.g., "confident" could refer

to the candidate or to the writer). The screening process left a list of 63 words as potential indicators of gender bias based on gendered usage and relevance to hiring. However, the direction and magnitude of bias were still unknown. We then stemmed the words on the list, replacing the original word with the entire "family" of semantically consistent words that shared a common root.

The final step identified letters containing one or more members of each word-family. Measuring multiple occurrences in a letter is necessary when counting lexical matches, as in Figures 1 and 2. A word list contains words (e.g., "achieve") that vary in meaning. Thus, a given letter might have multiple occurrences involving different list members with no repetition of the same word. In contrast, when counting individual words, a multiple occurrence is always a repetition of the same word.

*Lexicon words in a letter divided by total words (in %), averaged over letters, by category*

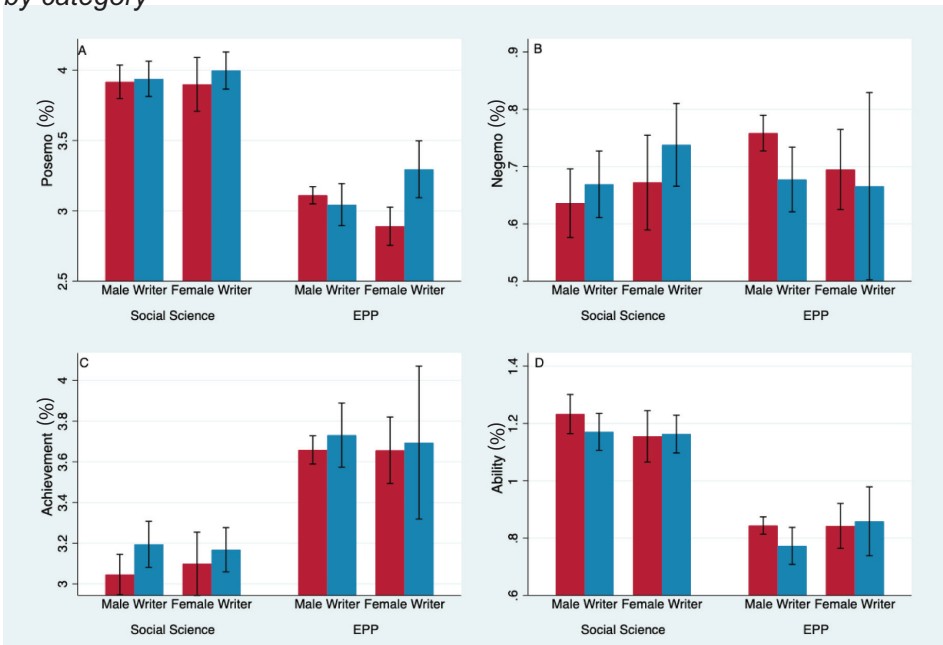

**Figure 1.** Four of eight lexical measures for male and female candidates, by discipline and gender of writer. The *y*-axis measures the percent of words in each letter that appear in the lexicon (e.g., "Posemo"), averaged over all letters in each of the four categories. Error bars are 95% confidence intervals. 842 letters were written by and for men in EPP and 301 were in social science; 176 letters were by men for women in EPP and 298 were in social science; 121 letters were by women for men in EPP and 139 were in social science; and 22 letters were by and for women in EPP and 307 were in social science. In EPP, women receive more positive affect words (**A**) than do men among letters written by women and fewer negative words (Panel **B**) among letters written by men. (Panel **C**) shows EPP uses more "achievement" words; (Panel **D**) shows social science uses more "ability" words.

*Lexicon words in a letter divided by total words (in %), averaged over letters, by category*

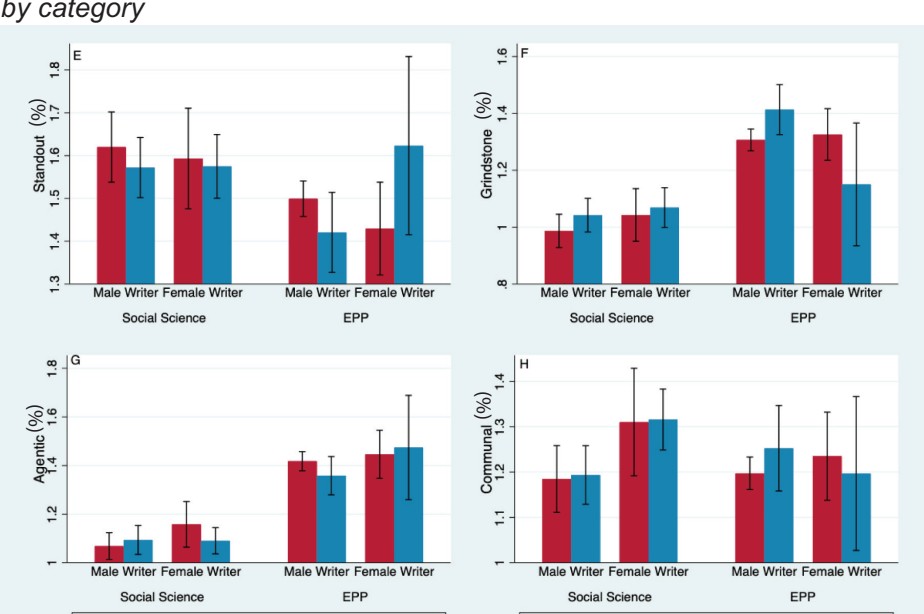

**Figure 2.** Four additional of eight lexical measures for male and female candidates, by discipline and gender of writer; these have been discussed in the literature and we present them separately. The *y*-axis measures the percent of words in each letter that appear in the lexicon (e.g., "Posemo"), averaged over all letters in each of the four categories. Error bars are 95% confidence intervals. 842 letters were written by and for men in EPP and 301 were in social science; 176 letters were by men for women in EPP and 298 were in social science; 121 letters were by women for men in EPP and 139 were in social science; and 22 letters were by and for women in EPP and 307 were in social science. (Panel **E**) shows the "standout" word rates. In EPP, writers use more "grindstone" words (Panel **F**) and more "agentic" words (Panel **G**). (Panel **H**) shows women use more "communal" words in social science.

For each letter, we removed punctuation, converted to lowercase and split text on whitespace to generate a sequence of words (tokens), excluding personal names and other identifying words. For the broad examination of terms used in letters, we limited our consideration to 7134 tokens that appeared in at least 10 letters and not more than 95% of the letters in the corpus. The reason for this limit is two-fold—words that appear in very few letters are likely to be identifying applicant or author and unlikely to represent systemic gender bias. Similarly, words that appear in almost all letters for candidates of both genders are unlikely to be associated with bias. These 7134 tokens constitute the base vocabulary for the open-ended analysis. We examined a subset of frequent words whose uncorrected *p*-values were significant at $\alpha = 0.95$. From the initial set of 646 words in our social sciences and 473 words in physics, we selected words with an average word frequency exceeding $10^{-4}$ in letters for at least one gender, yielding 218 words for social science and 159 for physics. Since the average letter length was about 1000 words, a frequency of $10^{-4}$ translates into an average of approximately once per ten letters. For each word in these subsets, we identified words describing a candidate attribute, a candidate accomplishment or a research area by inspecting randomly selected examples of the word from one of these categories that were dropped. The final labeled word counts were 116 gender-distinguishing words for social science and 97 gender-distinguishing words for EPP. The stem mappings are available in the Supplemental Materials.

### 2.3.1. Comment on Context

Word lists, of course, know nothing of context. It has been suggested that we search for statements such as "Compared to the very best postdoc physicists who have been in my lab, Candidate *X* is in the average range in terms of talent and accomplishment." or "Although Candidate *X* came to my lab with a very weak science background, they have subsequently gained considerable skill in designing studies and analyzing data." We have read all of the letters by hand. Statements like these (not unlike "doubt-raisers") occurred fewer than half-a-dozen times, referred to growth in the candidate, all for males, and would have made no noticeable difference in the results. There is of course inevitable subjectivity in this assessment, but our reading did not reveal any issues. In addition, words such as "weak" would have appeared in our "Negemo" measurement, which showed no effect.

We also discuss uses of the word "brilliant". Again, we checked by hand that the uses were not negated, such as "not brilliant" and similar phrases. This never occurred.

### 2.3.2. Fraction-of-Letters Analysis

Measuring gender disparities with individual words (instead of lists of related words) poses several methodological challenges. First, a noun might be more likely to be used in certain letters while its plural form might not, or a particular adjective might be used in those letters while its adverb form is not. For example, "superbly" is used in more social science letters for men ($p < 0.03$), but "superb" is not ($p < 0.98$), and when the counts are combined, there is no overall gender disparity. We therefore tested for gender differences in usage of a word's entire "family" of semantically consistent terms that share a common root, indicated by an asterisk (e.g., "superb*"). Semantic consistency was determined using WordNet (Miller 1995).

Deciding which gender-related terms are expressions of enthusiasm is inherently subjective. As expected, many gender differences reflect subfield specialization (for example, "family" in letters for female sociologists/psychologists and "neuroscience" in letters for men), and these terms were omitted from the analysis, along with gendered pronouns, function words, terms for which the gender difference was not statistically significant, and words with overtly context-dependent meaning ( "confident", for example, could refer to the candidate or to the writer). For the terms that remained, we used agreement between two of the senior authors, the physicist and one social scientist, who did not know the gender differences and did not communicate with each other about their choices. They independently identified 63 terms that were likely to express the writer's assessment of the candidate. For these 63 terms, we then included each term's entire family of semantically consistent variants that share a common root.

The probability that a particular word (or even word family) appears in a letter can be orders of magnitude lower than the probability that a long list contains a word that matches. Nearly all letters contain at least one match with a long word list like that used for "achieve", yet some of the individual words on that list might appear in only a few letters. We therefore used the larger of the asymptotic $p$ value (based on the $\chi^2$ test) and the exact $p_e$ value (based on Fisher's exact test) as the criterion for statistical significance. Fisher's exact test is more reliable for terms that are infrequently used, which is particularly important for EPP given the small number of letters written both by and for women ($N = 22$).

Usage frequency has a different meaning when based on individual words instead of word lists. Word lists contain large numbers of related words that nevertheless can have very different specific meanings. Thus, a given letter might have multiple occurrences involving different list members with no repetition of the same word. With individual words, multiple occurrence is always repetition. Repetitive usage by the same letter writer should not be assumed to be equivalent to single usage by multiple writers. For example, suppose letter A has zero mentions of the term "research," B has one mention, and C has two mentions. Given that "research" is mentioned one or more times in nearly every letter, there is a strong signal in writing a letter that never mentions "research." Thus, the difference between A and B is much more revealing than the difference between B and C.

Accordingly, we measured gender disparities as the percent of letters containing the term (or a member of the term's family), not the usage rate per letter.

Filtering terms for statistically significant gender differences risks false positives. We deliberately erred on the side of excluding terms with gender differences that were likely to be random, rather than including terms that were likely non-random. Each term had four opportunities to be statistically significant (gender of writer by word-level vs. letter-level differences), but the four opportunities were non-independent, which means that somewhere between 5% and 20% of the reported gender differences could be false positives. We were relatively unconcerned about false positives because our focus is not on gender differences for an individual term but on the pattern of differences at the disciplinary level. The gender difference (in the percent of letters) is equally likely to favor female and male candidates if the difference is random (i.e., a false positive). In sum, each individual gender difference should be interpreted as exploratory and potentially a false positive, but the deviation of the disciplinary distribution from gender balance can be interpreted as confirmatory.

### 2.4. Mathematical Methods and Non-Normal Distributions

We performed both non-bootstrapped and bootstrapped analyses for the lexical analyses (specifically a BCa bootstrap); no result quoted changed in the first three significant digits, indicating the results are stable to small changes in the analysis technique. We quote only the bootstrapped results. Since (a) we do not have a model or any useful prior for the either the distributions or the female/male differences we are investigating and (b) the distributions are not near boundaries, we used a frequentist bootstrap corresponding to a flat Bayesian prior. The fraction-of-letters analysis used non-bootstrapped statistical errors because a bootstrap analysis was not computationally practical. The open-ended analysis of Section 3.6 will use a permutation test to derive the significance of the signal from the data sample itself.

We have examined the robustness of our results to different analysis methods such as using medians instead of means to check sensitivity to non-Gaussian tails. Differences in means of distributions as reported here (and as reported in all of the literature we know of) do not differentiate between effects from the tails and shifts in the core or shape. We examine some of the raw distributions that show significant differences below.

## 3. Results

### 3.1. Standard Lexical Content

Given the striking underrepresentation of women in EPP, we were surprised not to find weaker letters for women. As discussed earlier, standout/grindstone, agentic/communal, positive/negative affect, and achievement/ability are eight categories used in past studies (Blue et al. 2018). These measures of letter content along letter length, and the rank and gender of the letter author revealed few statistically significant differences, with most favoring women. Figures 1 and 2 report the percent of the total words that matched each of these eight standard lexical measures, averaged across all letters in four groups based on discipline and letter-writer gender. Panels **A** and **B** show that (a) female physicists used more positive-affect words ($t = 2.41$, $p = 0.017$; $\Delta = 0.40\%$ words-per-letter, 95% CI: [0.65,2.6]%) and (b) male physicists used fewer negative-affect words ($t = 2.18$, $p = 0.030$; $\Delta = 0.08\%$ words-per-letter, 95% CI: [0.017,0.14]%) when writing about women than when writing about men. Consistent with previous studies, Panel **F** shows that male physicists used more "grindstone" words when writing about women ($t = 2.25$, $p = 0.025$; $\Delta = 0.11\%$ words-per-letter, 95% CI: [0.014,0.20]%). However, regardless of discipline or gender of writer, men were not depicted as more "agentic" or as "standouts," nor were women depicted as more "communal."

In social science, female writers used "communal" words (Panel **H**) more frequently than did male writers ($t = 3.31$, $p = 0.0009$; $\Delta = 0.09\%$ words-per-letter, 95% CI: [0.04,0.15]%), but they did this equally for male and female candidates in both the full sample and for the subset

of candidates with letters from both genders. Physicists used more grindstone, communal, and agentic terms than did our developmental social scientists, but female and male writers in EPP were equally likely to use these terms. The reader can see the size of the effects from Figures 1 and 2. Typically there are about 1000 words-per-letter; an effect size in the difference $\Delta = 0.1\%$ is then about one word.

### 3.2. Letter Length

Letter-length has been examined as a proxy for enthusiasm, as in Dutt et al. (2016). We could also ask if there is a difference between male and female writers across all candidates. There is in fact a large, significant effect. Figure 3 (Panel **A**) shows that letters for women candidates in social science were 65.4 words longer than letters for men ($t = 2.33, p = 0.02$; $\Delta = 65.4$ words, 95% CI: [11.0,120.])—more than 6%, which is somewhere between several sentences and a paragraph. In EPP, letter length did not significantly vary by candidate gender ($t = 0.37, p = 0.20$; $\Delta = 36.9$ words, 95% CI: [−38.4,112.]). Across both disciplines, women recommenders wrote longer letters than did men, a difference of 63.4 words ($t = 2.67, p = 0.008$; $\Delta = 63.4$ words, 95% CI: [18.,107.]). We are not the first to observe this phenomenon: a similar effect has been observed for surgical resident letters (French et al. 2019). This effect does not arise from tails, either high- or low-side. The core of the distribution is visibly shifted between male and female writers; Figure 4 displays some distributions to show representative differences.

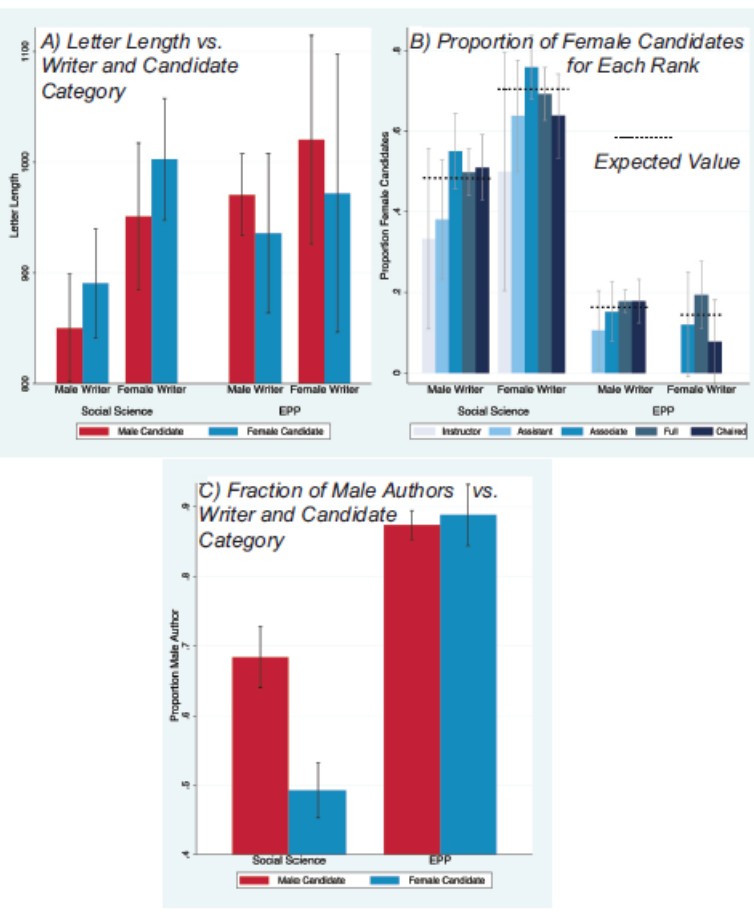

**Figure 3.** Word count (Panel **A**), proportion of letters for women across author academic ranks (Panel **B**), and male authorship of letters for male and female candidates, by discipline and gender of writer. Note the suppressed zero in the word count histogram. The dotted lines in Panel B show the expected values. In social science, letters for men are 19% more likely to be male-authored (Panel **C**). No other gender differences were statistically significant at $p = 0.05$. Error bars show the 95% confidence intervals.

*Sample Word Distributions*

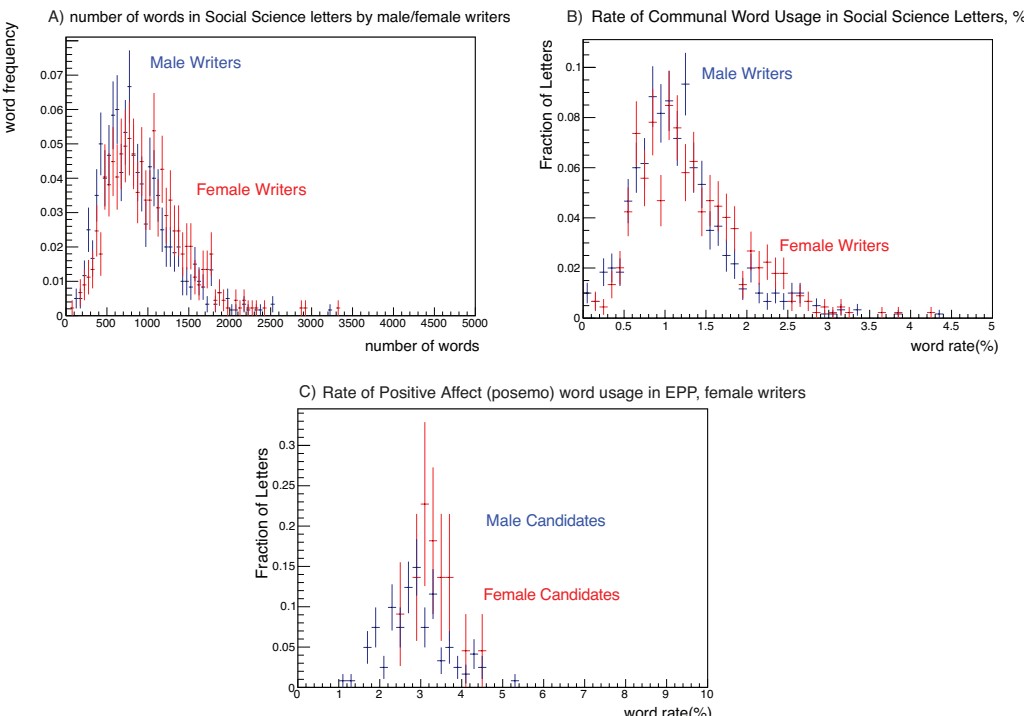

**Figure 4.** Three word distributions with significant male/female differences. (Panel **A**) shows the raw word count distribution for female and male writers in our social science sample. (Panel **B**) shows the communal distribution in social science letters for female and male writers; the graph shows that about 8% of the letters have a 1% communal usage. (Panel **C**) shows the positive affect distribution for female and male writers in EPP; the figure indicates about 15–20% of the letters have a 3% positive-affect word rate. All three pairs of distributions demonstrate the differences are not due to high- or low-side tails—we see the significant signals in the statistical tests of the text reflect actual differences in the underlying core distributions.

*3.3. Letter Authorship*

Figure 3 reports gender differences using two measures of the authorship of letters: the academic rank of the writer (Panel **B**) and author gender (Panel **C**). Academic rank was measured as one of non-tenure track instructor/lecturer, assistant, associate, full, and chaired professor or the rank equivalents. We observe no decline in the proportion of total letters written for female candidates as the rank of the author increases from instructor to chaired professor. This is confirmed using Spearman's rank-order correlation between the writer's academic rank and candidate gender; we found no significant correlations in either social science or EPP (the largest of the four correlations was very small ($\rho = 0.031$, $p = 0.45$) for male-authored letters in social science. The data also show that in both EPP and social science, male authors have higher academic rank, with a statistically significant difference in social science ($t = 3.07$, $p = 0.002$) but not in EPP.

If male writers have higher-status than female writers, we might expect a bias toward more male writers. Panel **C** reports differences in the gender of authors of letters for male and female candidates. There were no statistically significant differences in the gender of authorship in EPP ($t = 0.57$, $p = 0.571$) but the differences were surprisingly large in social science. Among all letters written for social science men, 68.4% were male-authored (301 male-authored vs. 139 female-authored), compared to 49.3% of letters for women (307 female-authored vs. 298 male-authored) with $t = 6.29$, $p < 0.001$. This "gender homophily" in letter authorship could reflect gender preferences of authors and/or candidates, or it

could be caused by gender differences in subfield specialization, or a combination of both, which we discuss in subsequent sections.

### 3.4. Differences Associated with the Sex of Writers

#### 3.4.1. Gender Homophily Effect

We noted that men in the developmental social sciences tended to receive letters from male writers. The existence of this gender homophily is independent of the actual ratio in the pool of possible writers: either the ratio is the same for male or female candidates or it is not. In order to estimate the size and direction of gender homophily effects we use our estimates of the writer pool both from demographic studies and our own estimates (available from the authors) in Table 3. Figure 3 (Panel **C**) and Table 2 show a large, significant discrepancy for the social sciences ($p < 0.001$) but none for EPP. Table 2 reports the expected F/M ratios among letter-writers for men and women—in EPP the ratio is 0.14 for both men and women, with an expected value of $0.16 \pm 0.022$ ($1\sigma$ statistical errors); in our social science disciplines, the ratios are $0.46 \pm 0.047$ for men and $1.03 \pm 0.084$ for women, with an expected value of $0.67 \pm 0.04$. The difference in the two means has a significance of $5.92\sigma$ with $p < 10^{-8}$. We stress that the expected value (with the source of the uncertainties described in the Supplemental Material) does not change the discrepancy between men and women applicants: a change in expectation only changes the relative degree of homophily assigned to male or female candidates. A far more extensive demographic study would be required to determine a more precise expectation, but the large and significant difference is established independent of that expectation. One possible and interesting explanation is that our two disciplines in the social sciences are partly gender-sorted by sub-discipline (see Table 4). Figure 5, discussed in more detail below, shows a possible sorting of terms associated with research on family and children appearing in letters about women candidates, with more "STEM"-associated terms appearing in letters for males. We stress that this is a post hoc analysis; we are not drawing a conclusion about the size of gender sorting into sub-discipline, or that the entire homophily effect we observe is explained by gender sorting. We are noting the obvious apparent correlation but stressing that any quantitative understanding will require a dedicated study beyond the scope of this paper.

**Table 3.** F/M ratios for the writer pools. The F/M ratio for potential letter-writers from the two different methods supplied in the text are (a) the first from available demographic sources, (b) from examination of representative departments. Uncertainties for the demographic methods cannot be obtained from the published data. See Appendix A for a discussion. We choose the method where we can explain the methodology and estimate the uncertainties as $1\sigma$ statistical (R.M.S./$\sqrt{N}$) in our sample.

| | F/M of Potential Writers | |
| | EPP | Social Science |
| --- | --- | --- |
| Literature Sources | 0.14 | 0.8 |
| Examination of Departments | $0.16 \pm 0.022$ | $0.67 \pm 0.04$ |

**Table 4.** Numbers of letters before and after requiring candidates have letters from both genders of writers.

| | Male Writer | | Female Writer | |
| Candidate | Male | Female | Male | Female |
| --- | --- | --- | --- | --- |
| EPP Total | 842 | 176 | 121 | 22 |
| EPP Both Genders | 302 | 47 | 121 | 22 |
| Social Science Total | 301 | 298 | 139 | 307 |
| Social Science Both Genders | 133 | 214 | 104 | 221 |

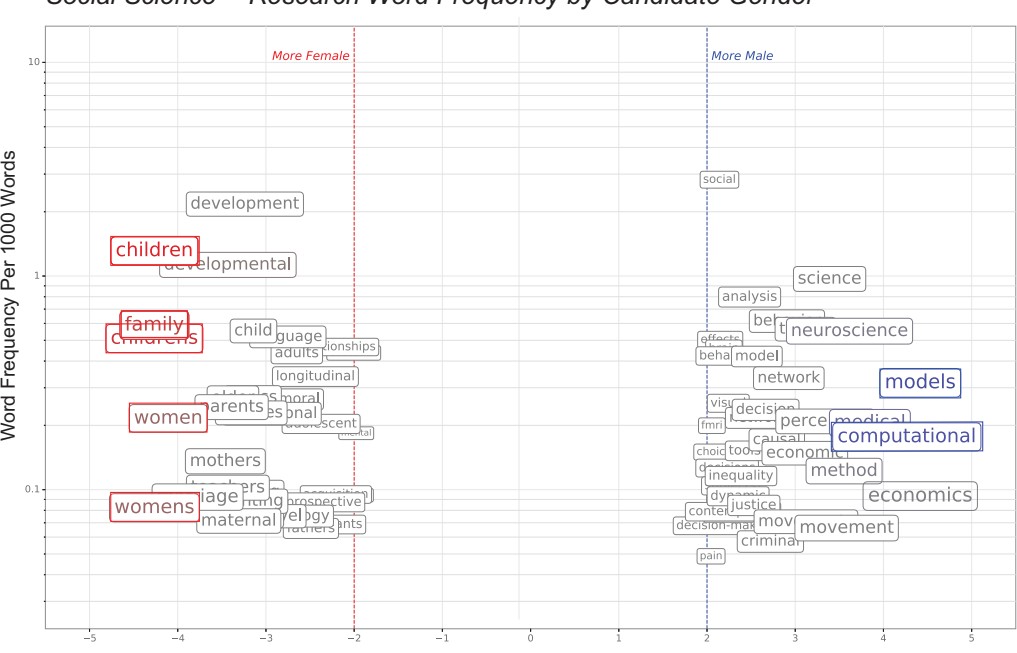

**Figure 5.** Research area terms, per-word, associated with candidate gender, in social science. Gendered research interests are apparent: note the clustering of words such as "child", "parent", "develop*" with women and "econom*", "comput*", or "model" with men. Some specific words such as "women/womens", "family*" and "children/childrens", in red, have Z-values −4 for women; "computational" and "models", in blue, have Z-values > +4 for men. The Z-score represents uncorrected *p*-values from the permutation test. The text color is a function of the corrected *p*-values which account for the expected false discovery rate when making multiple comparisons; any non-grey color indicates that the observed difference was among the most extreme values obtained in the permutation test but does not necessarily indicate significance). Words with $|Z| < 2$ are suppressed for clarity.

### 3.4.2. Differences in Letters by Male and Female Writers

We originally searched for differences focusing on candidates. As we analyzed the data, we discovered that there were significant differences associated with the sex of the author. Figures 1 and 2 report two significant sex differences in social science letters with no corresponding phenomena in EPP:

- Female social science writers write longer letters than male Social Science writers. The means differ; an analysis using medians instead of means shows a larger difference; and the distributions themselves are visibly shifted (see Figure 4), Panel **A**. A Kolmogorov-Smirnov test yields a $3 \times 10^{-8}$ probability the two distributions are identical.

- The rate of communal word usage by women authors in developmental social science letters was higher than the rate for men authors as shown in Figure 4, Panel **B**. A Kolmogorov-Smirnov test yields a $2 \times 10^{-7}$ probability the two distributions are identical. This is interesting in its own right, but combined with our gender homophily observation, earlier interpretations that an observation of higher usage of communal words for women candidates were due to gender bias should be reconsidered. If women tend to write for women candidates, and women writers use more communal terms, the finding that letters for women candidates have more communal terms might be correlated with gender homophily. Further research is required to disentangle these effects.

There is also a significant difference in the positive-affect use rate by female writers between male and female candidates, favoring female candidates, as shown in Figure 1 with the per-letter distribution shown in Figure 4, Panel **C**. Inspection of the letters reveals that positive-affect words such as "brilliant", "best", or "creative" and their variants account for the difference in the words-per-letter rate: $t = 6.0, p < 10^{-3}$; $\Delta = 0.31\%$, 95% CI: [0.20,0.42]%) and a Kolmogorov-Smirnov test for the per-letter rate gave a probability of $< 10^{-8}$ that the distributions were identical. The words-per-letter rates for both "communal" and positive-affect rates for male EPP authors were consistent.

### 3.5. Comment on Descriptions in Letters Not Captured by LIWC and Doubt-Raisers

We observed differences between the descriptions of EPP and social science candidates that are worth reporting but escape word counts. We performed tests for some of these words but the context, as with doubt-raisers, was such a strong confound that we chose not to report them. These differences refer to personal descriptions of the candidates. We give a few examples (quotes have been lightly edited to preserve confidentiality):

- A female writer compared a male candidate to Eeyore from Winnie-the-Pooh. Eeyore is often considered to be suffering from major depressive disorder, but the letter was otherwise favorable and in context the remark was meant as a compliment. Female writers described male candidates as "lovable", "charming", "delightful", "adorable", and similar words.
- A female writer said that a male candidate had "two adorable children" and used that assertion as evidence of the candidate's suitability for the position (but not because they had relevant experience with children or anything that related to research).
- A male writer said that a female candidate "demonstrated her ability to multitask through her raising of children while pursuing an academic career."

Quantifying this phenomenon would require another entire study, but we found (depending on our definition) 5–10% (50–100) of Social Science letters from women writers had such descriptions. There was precisely one mention of a candidate's children in the EPP letters, written about a female candidate by a non-US male writer, resembling the fourth instance above although the statement was not so direct. No other such personal descriptions occurred in EPP.

We distinguish such descriptions from cases where a writer said that "students love the candidate"—we are referring strictly to descriptions of the candidate by the writer involving personal characteristics or their family situation. Almost all such occurrences were by female writers but occurred when writing about both genders of candidates. The asymmetries in field and gender are striking: it would be unacceptable for a male EPP writer to say a female candidate was "lovable" or "adorable" but as we have found it is not uncommon for female social science writers to refer to male candidates with such terms. We also point out that endorsements of a candidate on the basis of having children may disadvantage candidates that do not have children.

Finally, we should explain our reasons for not including "doubt-raisers" in this study. It is routine in EPP letters to qualify the writer's knowledge of the candidate: examples include "I cannot assess their analysis skills" or "while I believe they may have been successful in hardware, I leave it to others to comment." Some writers in EPP have a standard paragraph attached to letters that are full of doubt-raisers and carefully explain that with the numbers of letters they write and post-docs they work with, they cannot know this particular candidate well enough to write a fully-informed letter covering all of the candidate's skills. These phrases are in context not intended to raise doubt but rather to be precise about the writer's knowledge of the candidate. Further, we found these qualifying remarks to be much more frequent in EPP than in social science. Rather than analyze a measure unsuited for our lexical analysis technique, subject to human interpretation, and used very differently in our two fields, we chose not to study the doubt-raisers category.

*3.6. Open-Ended Rate-Per-Word and Fraction-of-Letters Analyses*
Open-Ended Rate-Per-Word Analysis

Aside from words such as "she" or "him" that indicate gender, only a few words were significant after the false discovery rate correction. In physics, letters for female candidates were about three times more likely to include the term "brilliant" as a candidate descriptor. As a check, we also computed the number of letters containing the term "brilliant" and found that 28 of 963 letters for men and 16 of 198 letters for women contain the term, a difference in proportion with a *p*-value of 0.0011 ($\chi^2 = 10.68$, df = 1) favoring inclusion in women's letters (we checked the letters by hand and negations such as "not brilliant" never occurred.) This difference is particularly important in light of the findings reported by Leslie et al., who conducted a nationwide survey of academics in 30 disciplines and reported that gender asymmetries in obtaining PhDs is predicted by the extent to which academics in a field assume success in it requires native brilliance (Leslie et al. 2015). This has become a very influential explanation of the low numbers of women in fields such as EPP. The paper has (as of this writing) received 956 cites in Google Scholar since its publication; its Altimetric score of 1588 (as of this writing) places it well within the top 1% of media attention for publications in *Science* in 2015. The implication is that women in these fields are disadvantaged by gender stereotypes that associate men, but not women, with brilliance, as an explanation for women's underrepresentation in fields like EPP; the authors stated "The practitioners of disciplines that emphasize raw aptitude may doubt that women possess this sort of aptitude and may therefore exhibit biases against them". Thus, it is noteworthy that in our analysis of over 1000 letters written on behalf of actual job candidates in EPP, we found the use of the word "brilliant" to be more frequently found in letters about women candidates than men. We discuss this effect in greater detail in the open-ended analysis section.

For each word list category, we constructed scatter plots to check for gender differences that might not be otherwise apparent in the central tendencies reported in Figures 1 and 2. In physics, the only category word that is significantly gendered when accounting for the false discovery rate is the word "brilliant" from the LIWC "Posemo" (positive emotion) category (and "brilliant" is an "ability" word in previous studies (Schmader et al. 2007)). Our word count analysis showed no effect in the overall "ability" rate, which includes many similar words and dilutes the effect of any single word. We provide the raw letter-count rates for the word "brilliant" in Table 5. The reader can then make their own statistical tests for the significance of the use-rate in any candidate-writer gender combination. The data in other tables combined with these counts yield any rate information the data can provide, such as the rate of usage of "brilliant" by male writers for male or female candidates vs. its usage by female writers. The statistical power in this sample is low and meaningful comparisons are not simple; hence we present raw counts for the reader so they may make their own assessments.

**Table 5.** Count of letters with and without the term "brilliant." If a letter contains "brilliant" more than once, it is only counted once. Because of low statistics (3 and 4 uses of "brilliant" for female/female and male/female candidate/writer pairs) the reader needs to take care with drawing conclusions from this Table. Specifically, the data do not reveal any statistically significant difference in the use of the term "brilliant" for (1) male versus female writers or (2) male writers recommending male candidates versus female writers recommending female candidates.

| Candidate Gender | Writer Gender | "Brilliant" in Letter | Total Letters |
|---|---|---|---|
| Female | Female | 3 | 22 |
| Female | Male | 13 | 176 |
| Male | Female | 4 | 121 |
| Male | Male | 24 | 842 |

We next consider the open-ended detection of gendered words. We analyze words associated with candidate attributes separately in the fraction-of-letters section below. These categorizations suggested themselves from the data and were natural choices. We checked the letters after seeing words such as "students", "undergraduates", and "classroom" appear in our lists and learned they are associated with teaching accomplishments. The results appear in Figure 5. Research interests are associated with the most extreme *p*-values. In or social science disciplines, text for female candidates was two to three times more likely to include "children(s)" and "family" while text for male candidates was twice as likely to include the term "models" and four times as likely to include the term "computational". As noted, all the significant terms in social science related to candidate research area rather than a writer's comment about the personal traits of applicants. The remaining discussion also considers words that are not significantly gendered after the false discovery rate correction.

Words like "family", "children", "parent" and "development" are more female-associated (indicating women are more likely to perform research on families, children, and development than are men in these developmental social science areas) while "computational", "economic" and "network" are more male-associated. Research interests appear to be less gendered in EPP. The uncorrected *p*-values are mapped to standard *Z* scores to make the plots more intuitive. The plot text color is a function of the corrected *p*-values; any non-grey color indicates that the observed difference was among the most extreme values obtained in the permutation test. To orient the reader, the labels "More Male" and "More Female" appear at $\pm 2$ on the Z-score axis. The frequency corresponds to average word frequency for letters within the discipline.

There are several academic specific-words regarding publication, presentation, authorship, training and teaching that are associated with gender. Based on these observations, we recommend that future research on new data include a dictionary of domain-specific indicators of accomplishment. Our results are consistent with gendered research interests in developmental social science but it is entirely possible that these are all false discoveries. Our analysis only identifies the outliers, so our observations are possible even if the vast majority of research interests are not gendered. In sum, this internal validation provides compelling support for the main findings of what is essentially a gender-neutral landscape of letters of recommendation.

*3.7. Open-Ended Fraction-of-Letters Analysis*

The open-ended fraction-of-letters analysis revealed gender differences that were not evident in the lexical analyses in Figures 1 and 2 or in letter length and authorship in Figure 3. Gendered language favored women in social science, but in EPP, the disparity favored men. Of the 16 gender-differentiated terms of endorsement found in social science letters, 12 were more likely to be used to describe female candidates ($p = 0.027$), compared to only two out of eleven gender differentiated terms in EPP ($p = 0.028$).

Figure 6 reports the percent of letters in which the term appears. The *x*-axis displays terms with significant ($p < 0.05$) gender differences between candidates in the percent of male-authored and female-authored letters containing a term, based on all letters, regardless of the gender of the writer (due to insufficient statistical power among the small number of women writers in EPP). The symbol color is used to indicate whether the gender disparity is significant in letters authored by women only (red), men only (blue), or both (yellow). The open-ended analysis revealed gender differences that were not evident in the lexical analyses in Figures 1 and 2 or in letter length and authorship in Figure 3.

Within social science, contributions to knowledge (references to science and discovery and technical references) are more likely to be emphasized for men, while personal attributes (volunteering, delightfulness, initiative, leadership, ambition, accomplishment, success, and commitment) are emphasized for women. The largest disparities were in letters written by men. "Commit*" appears in 24.5% of letters about women, compared to 13.6% of letters about men, $p < 0.0008$), while "scienc*" is mentioned in 54.6% of letters for

men but only 45.6% of letters for women ($p < 0.03$). Of the 12 terms used more for women, only one, "volunteer*," is not used by male writers more than female writers. This reflects in part the greater statistical power for male-authored ($N = 600$) than female-authored ($N = 445$) letters. However, of the four terms used more for men, only one, "scienc*," is not used by female writers more than by male writers.

*Gender Difference in Percent of Letters Containing Term*

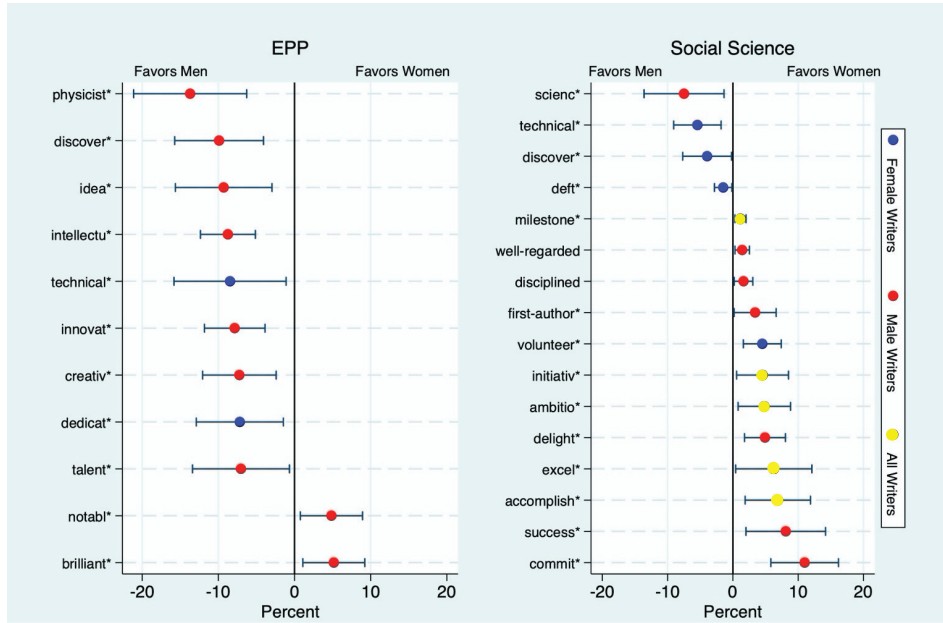

**Figure 6.** Gender difference (in percent) of recommendation letters containing a given term, by discipline. Error bars are 95% confidence intervals. We removed stop words, gender pronouns, references to gender differentiated sub-field specializations, and words with little or no gender difference in usage, leaving 405 words that were assessed by the senior physicist and a social scientist author, who agreed on 63 terms that signaled support for the candidate. These were expanded to 63 word families with a common stem and tested for gender differences in the percent of letters containing one or more occurrences of a family member. In EPP, nine terms were more likely to appear in letters for men compared to two for women. In social science, of 16 gender disparities, all but four favored women. Compared to letters about women, letters about men were more likely to contain references to science in social science and to physicists in EPP. In addition we note physicists used "brilliant" in nearly three times as many letters about women as men.

In contrast to the disparity favoring women candidates in social science, the gender disparity in EPP strongly favored male candidates whose letters were more likely to contain nine out of eleven gender-differentiated terms of endorsement. As in social science, "technical*" and "discover*" (as references to contributions to knowledge) were used more often for men, but unlike social science, male candidates were also more likely to be personally characterized as intellectual, creative, innovative, dedicated, and talented. Only two terms were used more for women: "notabl*" and "brilliant*." The latter is surprising, given recent findings that "brilliance" was more highly regarded in STEM disciplines with a lower representation of women (Leslie et al. 2015; Meyer et al. 2015). On the contrary, we found "brilliant*" was used in nearly three times as many letters about women (8.6%) than men (3.4%). A possible explanation is that letter writers may have been seeking to counter the unfavorable characterization of their discipline in response to the negative publicity from the prominent publication of this result. However, an informal email-based survey sent to a convenience sample of 74 particle physicists who routinely write letters revealed almost no awareness of this study. It is possible that physicists are aware of the signaling importance of "brilliant," even if they are unaware of the study, and believed that

designation was warranted in more letters written for women. Nearly all gender disparities in EPP were in letters by men, possibly due to very low statistical power for letters by and about women ($N = 22$). The two exceptions ("technical*" and "dedicat*") both favored men.

Comparing across the two disciplines, two of the largest gender disparities were for the terms "physicist*" in EPP and "scienc*" in social science. These two terms were used more often in letters for men than in letters for women (50.6% vs. 36.9% for "physicist*" ($p < 0.001$) and 61.6% vs. 55.0% for "scienc*" ($p < 0.03$). This could reflect—and possibly reinforce—unconscious gender stereotypes in which physicists and scientists are imagined as being men. That might not be surprising in a discipline that is overwhelmingly male, but it is noteworthy that the gender disparity for "scienc*" is one of the largest in a discipline that is majority female.

In sum, the open-ended analysis reveals a clear overall pattern of gender differences in the probability that a letter contains a term of endorsement, with the differences favoring men in EPP and women in social science. Moreover, Figure 5 omits four expressions whose widespread usage obscured gender differences in the proportion of letters containing one or more usages. In social science, references to research and teaching appear in 97% and 75% of all letters, respectively. References to research occur once per letter for female candidates and 0.90 times per letter for male candidates ($p < 0.008$) and references to teaching occur 0.40 times per female letter and 0.30 times per male letter ($p < 0.001$). Other differences in usage frequency ran counter to the overall pattern in Figure 5. In social science, "publish*" is used 0.12 times per female letter and 0.14 times per male letter ($p < 0.045$). In EPP, over half the letters contain references to "scienc*," with 0.2 usages per female letter and 0.15 per male letter ($p < 0.01$). Other terms in Figure 5 are far less common and do not indicate an overall pattern of gender bias in usage per letter.

The open-ended analysis could inform future LIWC studies. We could have used what we found in the open-ended analysis to create a specialized LIWC dictionary and reproduced the open-ended results; there would be no difference in the per-word rates, but the individual words could be lost in larger categories. Our hope is that future work will incorporate the techniques and words we found, but for this work we adhered to existing dictionaries as much as possible in order to see if we would find the same results. Finally, as we have pointed out, the per-letter analysis provides a different window into how specific, potentially "high-impact" words are used.

## 4. Discussion

Persistent underrepresentation of women in math-intensive academic disciplines like particle physics has been attributed to gender bias in recommendation letters that limit women's careers. We tested for gender bias in an important sub-field of physics and compared these tests to two subfields of the social sciences as a benchmark for what might be expected in a discipline with gender parity. Our results were surprising. In EPP, conventional lexical measures identified three significant differences, two of which favored women: letters for women candidates contained more positive affect in letters written by women and less negative affect in letters written by men. (Female physicists wrote fewer than 10% of the EPP letters, so less negative affect in the larger proportion of letters by men could be more consequential than the greater use of positive affect in the smaller proportion of EPP letters by women.) Male physicists also used more "grindstone" words when writing for women, which is consistent with previous studies and has been interpreted as a backhanded compliment based on the assumption that "effort" and "hard-working" are gender-biased code-words implying pedestrian research by women (Blue et al. 2018). However, Panel **F** in Figure 2 raises the possibility of an alternative interpretation. Male and female physicists used "grindstone" words twice as often as did social scientists, for all candidates regardless of gender, suggesting the possibility that these references have a positive connotation in usage by physicists. Across both disciplines, we did not observe differences found in previous studies showing that women are less likely to be depicted as

"agentic" or "standouts" (Madera et al. 2009; Schmader et al. 2007). In social science, lexical measures revealed no gender differences in letter content. Letter length favored women candidates, but the higher proportion of male-authored letters for male candidates could favor men, given the historical legacy of gendered status inequality. In sum, an exhaustive analysis of letter content using conventional lexical measures (Figures 1 and 2), as well as letter length and authorship in Figure 3, revealed gender disparities that, overall, were no greater in EPP than in social science, and a number of differences could be interpreted as representing an advantage for women over men.

Nevertheless, these results should not lead us to conclude that recommendation letters are free of gender bias. Lexical measures rely on lengthy pre-existing word lists that may include terms used more for women and other terms used more for men, with offsetting differences that could obscure the use of gendered language. For example, compared to men, women might have more descriptions saying they are creative but fewer saying they are innovative, with the aggregate score showing little or no gender difference. There may also be gender-differentiated words that are not in the word lists but which confer enthusiasm, as well as words that do not overtly express enthusiasm but may nevertheless indicate the use of gendered language.

We therefore supplemented the standard lexical measures with an open-ended search that checked every word in every letter for gender differences in usage, a methodology that has never been used in previous studies of gender bias in letters of recommendation. The analysis revealed gendered use of language favoring women candidates in social science (a discipline that is over 60% female), but favoring men candidates in physics (a discipline that is about 15% female). For example, men in physics were significantly more likely than women to be described as talented, intellectual, innovative, and creative, whereas women were more likely to be described as brilliant, a term that is more highly regarded in STEM disciplines with a lower representation of women (Leslie et al. 2015). Two of the largest gender disparities were for "physicist*" in EPP and "scienc*" in social science, both used more often in letters for men than for women. Although other explanations are possible, future research should investigate the possibility that recommendations can reflect unconscious gender stereotypes in which physicists and scientists are imagined as men, not only in a discipline that is overwhelmingly male but also in a discipline that is majority female.

There are limitations of the open-ended methodology as well. First, the interpretation of individual words depends on context, and even words that unambiguously express enthusiasm have unknown effects on hiring decisions. Our study reveals gender differences in how letters are written, not how they are read. We do not assume any disparity affected hiring decisions. Indeed, there were many more expressions of endorsement that were equally likely to be used in letters about women and men. Although Figure 6 includes terms like "brilliant*," "talent*," and "creativ*" whose stem variants are included in word lists used for Figures 1 and 2 and have been validated as exerting an influence on hiring, other terms (e.g., "delightful" and "physicist*") have not been shown to systematically influence hiring. Nevertheless, the gendered language is inconsistent with the assumption that letters for female candidates are indistinguishable from letters for males. In short, the lexical and open-ended methods have complementary strengths and limitations, which is why we adopted a multi-method approach.

All the methods we used share a common limitation: we cannot rule out the possibility that female candidates were superior to male candidates and deserved stronger letters than those they received. However, we controlled for candidate qualities when assessing gender differences between recommenders for candidates with letters from both genders. We also compared EPP with disciplines with gender parity as a benchmark. The similarity between EPP and our two social science disciplines in the number and magnitude of gender differences, despite a qualitative difference in the hiring of women, suggests that our null findings from the lexical analyses in Figures 1 and 2 might change very little were we able to completely control for candidate qualifications.

In addition, although our sample is much larger than those in previous analyses of gender differences in letters of recommendation, we may have underestimated the number of significant differences due to insufficient statistical power based on 1045 letters in psychology/sociology and 1161 in EPP. It is also possible that we overestimated the number of significant differences by using the conventional $p < 0.05$ benchmark for statistical significance. Figures 1–3 tested for gender differences on ten letter attributes (letter length, author rank, and eight different lexical dimensions), broken down by discipline and writer gender, plus author gender broken down by discipline. Out of 44 significance tests using a conventional benchmark of $p < 0.05$, we should expect two false positives if the null hypothesis were true. Had we performed Bonferroni-type corrections (called "look-elsewhere" effects in EPP) none of the observed lexical differences in Figures 1–3 would have been statistically significant. However, the differences associated with the sex of writers in our two social sciences, letter-length and "communal" rate, would have remained significant. It is also clear from the KS tests and the raw rate distributions of Figure 4 that the distributions are in fact significantly different.

The open-ended approach in Figure 6 is particularly vulnerable to false positives, and caution is therefore warranted in drawing inferences of gender bias in the use of any individual term. Of the 27 terms in Figure 6, only three gender differences would remain significant with $p < 0.001$: "physicist*" and "intellectu*" in EPP (both favoring men) and "commit*" in social science, favoring female candidates. The sign of the gender difference will be random if the difference is a false positive, since the measure is independent of gender differences in the number and length of letters. If all 27 terms in Figure 6 were false positives, an equal number would be expected to favor each sex. The probability that only two out of eleven random differences would favor women in EPP is 0.027, which supports rejection of the null hypothesis that the differences are false positives.

We summarize our reasons for not making corrections for false positives and using the $p < 0.05$ criteria: (1) given the persistent gender imbalance in many math-intensive disciplines, a false negative in tests for gender bias might be equally serious if it were used to justify existing practices that systematically favor male candidates and we therefore choose to risk over-reporting; (2) although $p < 0.05$ is an imperfect benchmark, using it as have previous studies makes possible a direct comparison with previous studies. A more stringent criterion, making Bonferroni-like corrections, would preclude such an "apples-to-apples" comparison.

Making the specific choices of EPP representing natural sciences and developmental psychology and sociology representing social sciences is just one reason to avoid over-generalization. It could be that EPP, a highly visible subfield of physics with large collaborations (thousands in some of the largest experiments) is different from other subfields with small groups of researchers; if this is true, the phenomenon may be worth further study. Caution is also needed when generalizing these results from entry-level to senior positions, from two high-profile institutions, and from National Laboratories to universities. At the same time, it is also important to note that candidates typically apply to dozens of positions, and letters for a given candidate rarely differ substantively from one search to another. Therefore, the letters in our samples are likely to resemble those submitted by these applicants to other searches well beyond these two institutions. It is also possible that there are confounds from the two particular institutions: perhaps the candidate pool for Fermilab and Cornell are different from other corresponding institutions. However, our study is not about the candidates, it is about the letters: one would have to hypothesize that the writers crafted their letters for the specific institutions in such a way to create or remove effects. We already know that EPP letters do not vary (aside from replacing institution names or other trivial changes). In fact, we often see "generic" letters that could be applied to any EPP institution without even name changes for a specific institution. In the social science sample, we again saw that the vast majority of letters were generic and many did not specifically refer to Cornell. We conclude such confounds (which would apply to the

foundational literature as well, which did not give the names of the institutions) do not affect our results.

It has also been suggested that we examine the outcomes of the searches. We did not do this for two reasons. First, the scope of this study was on the attributes of letters that might influence hiring decisions, not on how decision-makers respond to these attributes. Second, adding the outcomes brings in many non-lexical issues. Even if letters are written in an unbiased manner, they might still be read in a biased manner by search committees (Eaton et al. 2020; Moss-Racusin et al. 2012; Williams and Ceci 2015). Search committees can be influenced or even overruled by their institutions, so the decisions of a committee or the final hiring decision may not be a perfect measurement of the sentiments of the members. Removing such confounds is necessary in evaluating bias. These are not purely abstract possibilities: such effects unquestionably affected the selection process in the EPP searches. The EPP committees (multiple search committee members for typically three-year terms over the study period) agreed by consensus to use gender as a "tie-breaker" and internal discussions reflected an acknowledged bias favoring the selection of women candidates. The final selections among recommended candidates were made by Laboratory management, who had often not read the letters. We therefore do not examine the results of the search; this is a lexical analysis and there are too many other factors that affect the final decision that have nothing to do with the content of the letters. Future research should use randomized trials to manipulate the apparent candidate-gender of identical letters to test the possibility that search committee deliberations favor men in the evaluation of letters, despite the similarity of letters for women and men applicants. This was not possible in our study, since the size of EPP makes anonymity effectively impossible: randomizing names but otherwise leaving the letters the same (particularly the descriptions of a candidate's research) would not disguise the identity of the candidates. An entirely new study with a focused design is needed, well outside the scope of our research.

Future research on letter content should also include an open-ended search for all expressions of endorsement with gender disparities. Lexical analyses using word lists obtained from previous studies indicated gender disparities in EPP letters that were no greater than those observed in social science fields in which women are well-represented. However, our open-ended search of all gender-differentiated words revealed terms of endorsement that were used more often in EPP to depict men and in social science to depict women. Taken together, these two methodological approaches spotlight the need for future research exploring gender bias as well as other causes of the shortage of women pursuing academic careers in physics. Policies to correct gender imbalances in math-intensive fields may be more effective if they target barriers in addition to bias in letters of recommendation, such as how letters are evaluated, and, most importantly, obstacles that discourage women from choosing math-intensive academic careers.

**Supplementary Materials:** The following supporting information can be downloaded at https://www.mdpi.com/article/10.3390/socsci11020074/s1: 1. Master spreadsheet for LIWC analysis: scienceMasterSpreadsheet.xlsx: Contains all results from the LIWC analysis of letters. Includes both LIWC2015 words and non-LIWC2015 words gathered from the literature and then used in the master spreadsheet. 2. Dictionary of words not in the LIWC 2015 Dictionary: extraCategoriesDict.dic. 3. Stem mappings used in the per-letter analysis: word_mappings_fig_3_final_v12.csv: These are the stem mappings used in the preparation of Figure 6. 4. STATA code used in *t*-tests: ttestsForPaper.do. 5. STATA data for per-letter analysis in social science: soc_masterfile_stemmed2.dta. 6. STATA data for per-letter analysis in EPP: epp_masterfile_stemmed2.dta. 7. STATA code for the per-letter analysis used to create Figure 6: fig3_2color.do. 8. Survey of gender ratios of departments writing letters that are near Cornell in US News and World Report rankings: sociologyDepartments.xlsx. 9. Our survey of the gender composition of institutions that wrote many of Fermilab's EPP letters: physicsDepartments.xlsx.

**Author Contributions:** R.H.B., S.J.C., M.W.M., C.J.C., and W.M.W. designed the research; R.H.B., M.W.M., and C.J.C. analyzed data; R.H.B., C.J.C., and S.C.W.-C. coded data and organized/maintained

spreadsheets; R.H.B., M.W.M., W.M.W., S.C.W.-C., C.J.C., and S.J.C. wrote the paper. All authors have read and agreed to the published version of the manuscript.

**Funding:** M.J.M. received National Science Foundation funding from SBR 2049207 and SBR 1756822.

**Institutional Review Board Statement:** Exemption from IRB Review has been approved according to Cornell IRB Policy #2 and under paragraph(s) 4 of the Department of Health and Human Services Code of Federal Regulations 45CFR 46.101(b).

**Data Availability Statement:** Since the relatively small size of EPP makes it effectively impossible to anonymize the original letters, those (for both EPP and social science) will not be made available. All other results and code necessary to reconstruct results in this paper are at https://www.mdpi.com/article/10.3390/socsci11020074/s1 , or can be made available by the authors without undue delay on reasonable request.

**Acknowledgments:** We thank Fermi National Accelerator Laboratory (Fermilab) and Cornell University for use of the letters. We would also like to thank the developer of LIWC, James W. Pennebaker of the University of Texas, Austin, for valuable discussions.

**Conflicts of Interest:** The authors declare no conflict of interest.

## Appendix A. Female-to-Male Writer Ratio

Domain knowledge is essential for making estimates of the expected F/M pool for writers. We noticed that letters came from Assistant Professors less frequently and we were concerned that the gender ratio might be a function of academic rank (for a fuller discussion see Section 3.3), so we attempted where possible to only use tenured faculty. We followed two methods for each field (the raw data for making estimates from the data are available from the authors and the Supplemental Materials for this article):

**EPP Measurement from Data** In EPP we counted the number of Associate or Full Professors at representative schools and Laboratories during the period in which the letters were written. We examined the web pages of each department; EPP faculty are readily identifiable. We obtained a value of F(F+M) = 0.14, or F/M = 0.16 ± 0.022. An older list from the American Physical Society's Divison of Particles and Fields membership file (in 2014) was examined for potential letter-writers with the same criterion for academic rank and gave 0.16 ± 0.03; that list is confidential. These values are consistent indicating that we are making a reasonable estimate for our EPP writer pool.

**EPP Demographic Studies** In EPP, we first used data from the American Institute for Physics (Porter and Ivie 2019). These data do not break out the fraction of Assistant, Associate, or Full Professors and report 0.14 for all of EPP in 2017. We checked this estimate against The National Center for Science and Engineering Statistics (National Science Foundation 2019) (NSF 19-301, Table 16, 2018), which gives 0.158 for all of EPP in 2017. Uncertainties are not quoted for either survey.

**Social Science Measurement from Data** We examined a total of 26 sociology and psychology departments both above and below Cornell in US News and World Report rankings (as a convenient proxy for schools at the same level). For each school, we examined the web pages of each sociology and psychology department and counted Associate and Full Professors, leaving out Assistant Professors, lecturers, adjuncts, and emeriti. The gender of each person was determined from names or pictures. This method was problematic: the definition of faculty in appropriate departments is not uniform and we found the information available on the Internet to be incomplete, out-of-date, or not usefully organized. It could also be that applicants are either from non-psychology departments (education, human development, brain sciences, information sciences, communication departments, medical schools, law schools, or business schools) or worked with researchers from non-psychology departments, possibly biasing a count based on the scan of just sociology and psychology departments.

The definition of the relevant department also varied across schools: for example, Cornell's Department of Human Development has neither sociology nor psychology in its name but is one of the organizational units to which candidates applied. Estimating the expected social science ratio was therefore much more difficult than in EPP. We nonetheless followed this procedure and stopped after (M = 328, F=229, F/M = 0.70) in Sociology and (M = 469, F = 300, F/M = 0.65) in psychology since the results were not changing with additional statistics and the (unestimated) systematic errors of the method made it pointless to continue. The sum over both departments is for F/M is $0.67 \pm 0.04$ (statistical errors only). The raw data are available from the authors or at supplementary.

**Social Science Demographic Studies** We again used NSF's 2017 Survey of Doctoral Recipients (National Science Foundation 2019). The NSF 2017 Survey of College Graduates tells us the ratio F/(F+M) in 2010 for self-identified sociologist/anthropologists, other social scientists, and secondary teachers was 56%. Although 53% of assistant professors are female in NSF's national data during the time frame of this study (2010–2017), only around 45% of tenured faculty are female, and as we noted letters were written less often by Assistant Professors. However, the 45% female figure is based on the number of female tenured professors at ALL colleges and universities, not the universities that comprised the bulk of letter writers for this study. More web searches told us there are disproportionately more females at small 2-year and 4-year teaching colleges and more males at the R1s. Furthermore our writers came from more than just psychology departments: we had writers from brain science departments, human development departments, medical schools, business schools, communication departments, law schools and more. A further problem revealed itself at Cornell itself: 10–15% of the faculty in Cornell's psychology and human development departments have degrees outside psychology in such diverse fields as electrical engineering, political science, biology, and sociology.

We see from Table 3 that the EPP estimate is consistent across the two methods (examining representative institutions and the demographic surveys) and the social science value is consistent as well, even though no uncertainty has been assigned to the demographic methods. We quoted the demographic methods but since there is consistency and no result is dependent on the precise value the issue is not problematic for this work.

Is there some effect that would cause the bias measurement to be dependent on our particular institutions? As mentioned earlier, letters for candidates are routinely "recycled" in EPP for all positions, so there will be nothing particular about Fermilab's letters. Many of the letters we saw for Cornell do not even mention the institution, so the letters were not crafted with Cornell in mind. Candidates do make choices about the institutions to which they apply, but this study is about the *letters*. Candidates may well select particular institutions, but to create an effect in our data one would then have to hypothesize the letters differ across institutions in ways that introduce or remove bias—but we know from the letters that they are used across many institutions. Hence there is no reason to believe our letters are somehow atypical.

**Appendix B. Comparison of Current and Former Studies**

How do the findings of the current study compare with those from prior studies? Here we examine all previous studies of academic letters, some of which have reported divergent results from the current study, which as was seen, found few gender differences, most of which favored women. In this section we highlight four possible reasons for any inconsistencies between these results and former ones.

1. Date: We used an open-ended analysis that reversed the word count methodology to provide an internal validation of the word count analyses. Instead of starting with "recommendation" words in published word counts and then measuring their gender frequency, we started with words with gendered frequency and looked for those

indicating strength of recommendation. This is an independent analysis that is not dependent on the words in word counts used in the main analysis but rather is based on the gendered associations of all words in actual letters. Between 30% and 86% of the patterns in each word count category matched a word in the letter text, but over 75% of the words in the base vocabulary considered below were not found any word count. Therefore because a great majority of the words in the letters are not in the word count categories, it is possible for large differences in the letters to be overlooked. We examined three variations of mapping letters to sets of tokens (1) within word-category word use differences based on rate per word frequencies, (2) a broad examination of all common words based on rate per word frequencies, and (3) an examination of candidate descriptors based on fraction-of-letters frequencies.

2.  Different Fields: Some previous studies examined applicants for jobs in fields that are less math-intensive and have much greater female representations than EPP (for example, medicine, biology, and geoscience). *It is therefore even more curious that some of these studies found gender bias in letters of recommendation, while we found little evidence of gender bias in EPP.* We therefore urge extreme caution in generalizing our results beyond EPP and social science, including even other subfields of physics such as theoretical particle physics. The core values for a field such as EPP, which requires large collaborative efforts (hence, a possibly greater emphasis on communal and grindstone traits), may differ from fields in which research is carried out by much smaller collaborative teams or even by lone individuals. Again, the social sciences are usually less collaborative and math-intensive than EPP and have dramatically higher representations of women than EPP so we expected the social science effects if anything to be markedly smaller, which they were not.

3.  Nationality of Writers: Some prior findings, particularly Dutt et al. (2016), that are inconsistent with the current findings contained a large proportion of letters written by recommenders from outside America. That study finds non-US letters are much shorter and they are less enthusiastic than letters written by American scientists. Thus, there is evidence that differences in letter length and tone vary significantly across cultures, with letters written by Americans being longer and more positive than those emanating from other regions. The vast majority of the letters in the present study were written by Americans; in contrast, in Dutt et al. (2016), 530 out of 1224 letters were written by scientists outside the US. Thus, the current study clarifies what (until now) has been a set of seemingly contradictory findings that result from studies that are mostly small-scale and based on samples lacking important controls (such as the unavailability of letters for unsuccessful applicants, or too small subsamples of female writers to do gendered analyses). Some, but not all, of these earlier studies have reported evidence of gender bias; however, there are many exceptions and contradictions that we enumerate next.

4.  Sample Size: The current study is based on the largest sample of letters thus far examined, 2206 letters. Prior studies varied from 237 letters in Li et al. (2017) to 1224 letters in Dutt et al. (2016). This is important because the smaller samples precluded examining correlations between the gender of applicant and the gender of the writer. As noted elsewhere in this article, it is important to remember that candidates typically apply to many positions and letters for a given candidate rarely differ substantively from one search to another. Therefore, the letters in our samples are likely to resemble those submitted by these applicants to other searches beyond these two institutions. Nonetheless, caution is needed when generalizing these results; for example, the internal culture of EPP, with large collaborations requiring communal behavior, may differ from other fields of physics.

5.  Dependent Variables: With the exception of McCarthy and Goffin (2001) the current study employs the largest number of dependent variables: length, agentic, communal, grindstone, standout, achievement, positive affect, negative affect, ability, homophily, and writer status. The current study also included an internal validation using words

not limited to the word counts used by prior researchers, an analysis that was bottom-up. This analysis began with words associated with gender that were not contained in any of the word counts; that is, after finding few gender differences in the myriad dependent measures examined, the current study undertook an independent test that confirmed no other terms associated with gender were related to ability; instead, they were words that described research topics that tended to be gendered (as perhaps implied our findings, women in developmental social science being more likely to study family processes and men to study neuroscience.)

6. Control of Background: Only a few prior studies attempted to control for applicants' personal characteristics (by using number of publications, conference presentations, class rank, and/or awards as covariates). These are less than ideal controls for other ways applicants can differ (e.g., status of their mentor, quality of the journals in which they publish, prestige of their university, their contribution to multi-authored studies). In contrast, the current study included an analysis of a subsample of 918 of the 2206 letters for candidates with letters from both genders, neither of whom was the candidates primary advisor. Each candidate thereby contributed to the measures for each writer gender. This minimizes differences between male and female candidates caused by gender differences in applicants' personal characteristics. However, it does not address the possibility that differences in letters for male and female candidates might reflect unmeasured differences in candidate qualifications.

7. Comparing Disciplines that Differ in Women's Representations: The current study contrasted two disciplines in which women are disparately represented. Only Dutt et al. (2016) examined a math-intensive field, the analysis of letters for postdoctoral positions in geoscience. This current study examined an even more male-dominated field, EPP (Elementary Particle Physics) and contrasted it with fields with high female representation, psychology and sociology. This contrast provided a principled basis for the expectation of larger correlations between gender and the dependent variables within the less female-represented field, EPP.

In sum, the current study was much larger and contained more measures than in most former studies.

Trix and Psenka (2003) analyzed 300 letters written on behalf of applicants for faculty positions at a single U.S. medical school. The letters were written in the mid-1990s, preceding the secular movements that occurred two decades later. Unfortunately, Trix and Psenka only had access to letters written on behalf of successful applicants, precluding a comparison with unsuccessful letters. They also were unable to analyze their data as a function of letter writer's gender, due to their small sample, rendering their results more narrative than quantitative. Like the current study, they found no differences in the frequency of letters that contained standout terms (58% for men vs. 63% for women). However, they found that letters written for men tended to repeat standout terms: on average, such letters contained 2.0 standout terms vs. only 1.5 for women's letters. Because of the unavailability of letters for unsuccessful candidates, there is no way of knowing how instrumental these features were in hiring decisions. Trix and Psenka also found that letters for women contained twice as many "doubt-raisers" as did letters for men. Finally, like Dutt et al. (2016) they found that letters by writers from Europe were shorter: "Even letters from Canada were less hyperbolic than those from the USA. But we did not have enough letters to make more than general observations." Thus, Trix and Psenka's study was limited by its small sample size, a single institution, and a single field (medicine), and the authors were unable to analyze letters for unsuccessful candidates or as a function of the gender of the writer, precluding many of the analyses in the current study. Despite these limitations, Trix and Psenka provided a rich narrative analysis that influenced the hypotheses in the current study.

Messner and Shimahara (2008)'s study was twice as large as Trix and Psenka's, 763 letters vs. 300 letters, written on behalf of applicants for a 1-year residency in otolaryngology/head and neck surgery at Stanford University's Medical School. Only 8.8% of

letters were written by women, which limited gender of applicant times gender of writer analyses. They found that all letters were quite positive, which echoes Dutt et al. (2016)'s finding that roughly 98.5% of letters were either good or excellent. However, Messner and Shimahara (2008) found that letters written for women contained more communal terms (e.g., team player, compassionate), and male writers were more likely to mention a female applicant's physical appearance. We found zero such occurrences in EPP, although there were statements in social science letters by female writers that male candidates were "adorable" or "charming" as discussed earlier. They found that 86% of all letters contained standout terms (averaging 2.6 per letter). However, like the current findings—and unlike Trix and Psenka's—standout terms did not differ by gender of applicant. They also found that doubt raisers (present in 19% of letters) did not differ by gender of applicant, unlike Madera et al. (2009) and Trix and Psenka (2003), both of which reported more "doubt-raisers" for women applicants. (Trix and Psenka 2003) Finally, unlike most studies (e.g., Trix and Psenka (2003))'s), Messner and Shimahara (2008) did not find a difference in letter length as a function of gender of writer or gender of applicant, nor did they find a correlation between letter length and favorability. However, the mean length of their letters was less than half of the length in the current study: female writers' letters = 345 words, male letter writers' letters = 328 words (not statistically significant); in contrast, the mean length of letters in the current study ranged between 915–960 words, which is considerably longer than letters written in other studies. Our much longer letters will be qualitatively different, with more depth and detail. Comparisons between these two sets are then complicated by the evident difference in the commitment of the writer.

In a larger study of letters written for geoscience postdocs, Dutt et al. (2016) analyzed 1224 recommendation letters, submitted by writers from 54 countries (43% were from outside the U.S.), for postdoctoral fellowships in a single field, geosciences, submitted between 2007 and 2012. (Dutt et al. 2016) Unlike Messner and Shimahara (2008), Dutt et al. (2016) and her colleagues found that letters written for women contained fewer words. However, like both Messner and Shimahara (2008) as well as the current study—but unlike Trix and Psenka (2003)—Dutt et al. (2016)'s letters contained similar numbers of standout words and more grindstone words. Although these researchers found that female applicants were only half as likely as men to receive excellent letters, they found no evidence that male and female recommenders differed in their likelihood to write stronger letters for male applicants. Like Trix and Psenka (2003), they also found that letters from American writers were on average 561 words whereas those written by Africans, (305 words), South Asians (275 words) and East Asians (320 words) were notably shorter; even Europeans, New Zealanders and Australians wrote shorter letters (345 words) than American writers. In contrast to the current findings, Dutt et al. (2016) concluded: "these results suggest that women are significantly less likely to receive excellent recommendation letters than their male counterparts at a critical juncture in their career."

Madera et al. (2009) analyzed 624 letters from an earlier study that had been written for 174 applicants who had applied for positions in academic psychology at a single R1 university. Letters written for females contained more doubt-raisers, even after controlling for personal accomplishments (number of first-authored publications, honors, etc.) In this regard, their findings agreed with several of the above studies such as Trix and Psenka (2003) but disagreed with several others.

Li et al. (2017) analyzed 237 letters written on behalf of applicants to a four-year emergency medicine residency at Northwestern University. (Li et al. 2017) Of the fifteen dimensions they analyzed, only three revealed gender differences: letters written for female applicants were slightly longer, contained more ability terms that referred to expertise, competence, and intelligence, and also more affiliative/communal terms that referred to teamwork, helpfulness, communication, compassion, and empathy. Unlike other studies such as Trix and Psenka (2003) they found no gender differences in doubt-raisers, grindstones, or standouts. Overall, they found little evidence of gender bias in letters, although the special nature of the application process may have influenced their findings (including

the constraints that letters were limited to 250 words in length and only the top quarter of applicants were invited to apply to apply. In contrast, French et al. (2019) found that letters written by females for surgical residents were 45 words longer than letters written by males for both genders of applicants) and also that writers used standout terms more for letters about female applicants than male applicants.

Schmader et al. (2007) examined 886 letters written for 235 male and 42 female applicants for a chemistry/biochemistry faculty position at a single R1 university. Sample sizes were too small to analyze data by gender of writer times gender of applicant, so many of the analyses in the current study were not possible. The word count of letters for women was 604 words vs. 555 for men. They found no gender differences in the frequency of grindstone words. However, unlike the current study and several others, recommenders used significantly more standout adjectives to describe male than female applicants. Letters containing more standout words also included fewer grindstone words, which runs counter to the current study's finding of weak statistical relationship ($r = -0.04$) between the co-occurrence of grindstones and standouts.

Interestingly, in the earlier analysis, Madera et al. (2009) analyzed 624 letters written for 194 applicants in psychology and found that male recommenders wrote 262 letters for male applicants and 194 letters for female applicants; in contrast, female recommenders wrote 78 letters for male applicants and 109 letters for females. Hence, to some extent they resemble the current study's finding that male applicants submit more letters from male writers than from female writers and the reverse trend for females, which the current study did not find (that study found homophily only for males in the social sciences). Madera et al. (2009) also found that women were described as more communal and less agentic than men, neither of which was found in the current study. Finally, although they found more agentic adjectives in letters for males, there was no difference in "agentic orientation", a summary of indices of how much writers referred to the applicants as active, dynamic, and achievers (using words such as "earn", "insight", "think", "know", and "do").

The study of recommendation letters has continued to the present: we examine two we found of particular relevance. Powers et al. (2020) studied a larger sample than ours for an orthopaedic residency program in 2018, examining 2625 letters for both race and gender. The reference letters were standardized to reduce potential bias, a relatively new idea. The researchers concluded (UiO indicating "underrepresented in orthopaedics"):

> Small differences were found in the categories of words used to describe male and female candidates and white and UiO candidates. These differences were not present in the standardized LOR compared with traditional LOR. It is possible that the use of standardized LOR may reduce gender- and race-based bias in the narrative assessment of applicants.

The study was performed using LIWC 2015 for the standard categories: agentic/communal, standout/grindstone, and ability. Interestingly, the researchers concluded that standardized letters of reference may only produce a small effect. They also made an interesting speculation that a orthopaedics-focused word list may have obscured bias; our reverse methodology addresses this issue and is a powerful method for going beyond LIWC or other pre-defined lists.

These authors also note:

> A similar discrepancy has been noted in studies analyzing letters of recommendation for surgical residency and suggests that applicants preferentially ask men faculty over women faculty for letters of recommendation... If applicants believe that letters from writers of higher academic rank carry more weight, then the larger proportion of men at higher academic rank could be one explanation for this difference...

The former is exactly what we have observed in our developmental social science letters, although in our study we noted the clustering into research areas as in Figure 5 with

no corresponding effect in EPP. These authors also hypothesized a rank/weight correlation, but we observed no significant effect.

Kobayashi et al. (2020) studied 2834 letters in another orthopaedic residency study. Their conclusions, again using LIWC 2015, were quite similar to ours:

> Although there were some minor differences favoring women, language in letters of recommendation to an academic orthopaedic surgery residency program were overall similar between men and women applicants... Given the similarity in language between men and women applicants, increasing women applicants may be a more important factor in addressing the gender gap in orthopaedics.

The authors made some of the same points regarding the limitations of word lists made in Powers et al. (2020) and in the current work.

To recap, amidst many similar findings, there were also many differences between the current study and former studies, any of which might be responsible for inconsistencies when they occurred. There are no studies directly comparable with the current study: they are either older (predating the recent focus on gender issues in STEM), not written on behalf of applicants for tenure-equivalent positions in STEM fields, and/or written for less math-intensive STEM fields that have higher representations of women, or written by writers from different cultures. All but one, Li et al. (2017), employed fewer measures than the current study. These differences may partly explain why we found less evidence of gender bias against women candidates than might have been expected from statements made in articles such as the following quote from Blue et al. (2018):

> Standout words in letters of recommendation...portray a candidate as talented and exciting, (and) are most often found in letters of recommendation for men. Grindstone words create the impression that a candidate works hard but is not intellectually exceptional, (and) are more often used for women...As a result of that discrepancy, female candidates seem both more boring and less intellectually promising than their male competitors.

This article appeared in Physics Today, a magazine for members of the American Physical Society and was not based on original research by the authors; it was written for physicists who wanted to understand the general issue and accurately reflected a distillation of common sentiment.

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
