# Peer review of "Assessing Gender Bias in Particle Physics and Social Science Recommendations for Academic Jobs"

_socsci, doi:10.3390/socsci11020074_

Round 1
Reviewer 1 Report
The authors have done a systematic study. Discussion has been thoroughly documented. They manage to erase a higher sensitivity for the issue.
Author Response
Response to Reviewer 1:
Thank you for your careful reading. We did not see anything that you wanted us to change.
Reviewer 2 Report
This is a thoroughly researched and well-written paper, making a genuine (and well-spelled-out) contribution to the existing literature on the topic. It is good to see repeated throughout the point that this paper is about the letters because many of the thoughts it provokes (in the Reich-Ranicki sense) relate to associated issues that require separate treatment … one example is the relationship between the candidates, interview panels and the letters because I would hazard in the real world many would be either unread or cherry picked for attention because of perceptions of the authors of the letters, which this paper does touch on tangentially and is methodologically partly implicit in the decision to prioritise faculty writers. That this paper provokes wider thinking is a strength.
Very good coverage of prior published research. Appendix B provides a reasonably thorough literature review and the body of the paper is well and carefully referenced.
Excellent introduction, concluding with an absolutely key observation (lines 48-49). The second paragraph of the introduction points to a core difference between the gender/sex distribution of the available pool of candidates between the two fields. In general, there is a demographic structural difference also between these two pools of candidates: the social science candidates are likely to be older. I may have missed it but I didn’t see a reference to age in the suppmat. This could imply a difference in the choice of referee which sits alongside the gender issue, and which may also nuance the perceptions of a candidate’s ability. It would also be expected that the mindset embedded in, and associated with, the fields may have an impact …. terse precision in EPP, greater prolixity in DSS, for example. The latter is, as the paper acknowledges, a much more diverse and broader field that EPP – this becomes important for the discussion on letter length in section 3.2.
Section 1.1.1 describes what this paper brings to the debate well. Point 1: I would question whether either psychology or sociology within the developmental social sciences in fact reflects gender-parity (L87).
More pertinent perhaps to the topic here is the later discussion on gender homophily effect, pointed out in 1.1.2. . Point 2 (LL90-104): correct approach but a side question: would either the gender of the primary advisors or their relative weight/reputation make a difference? And was there a difference between candidates applying within schools as opposed to external to schools? (may influence choice of referee writers) [in part this is covered in 1.1.2 point3.]
Methods are well described and include a thorough discussion in known limitations in the approach.
Results thoroughly described. One question that jumps out: was analysis done looking at differences in letter content of successful candidates and unsuccessful candidates?
The discussion on syntactic differences (2.3.2 para 1) and the effect on semantic analysis is a key topic that might reward further analysis at some stage. One observation related to lack of difference in EPP (lines 380-382) relates also to the point above: EPP writers are much more likely to be formulaic than DSS writers, which, if put alongside issues described in 3.4.2, would suggest the difference observed here would be expected.
Figure 2: is there a missing “of”? (four additional of eight)
Section 3.5 is in many ways the most important section for three reasons: it identifies potential limitations in the analytical power of the techniques applied here, it identifies a major difference between fields and it points to an area of future research. It also suggestions a question, also suggested by the conclusion, that there may be differences in the relationship between the writer and expectations of the field of study/employment on the one hand and the relationship between the writer and the candidate on the other which may have a bearing on the context and outcomes.
Line 588: is this supposed to be “scienc*” or, as on page 19 “scien*”?
Does the paper achieve what it sets out to do as described in 1.1.2? The short answer is yes. Does it point to pathways for future research and remaining gaps? Yes.
Author Response
Response to Reviewer 2:
Section 1.1.1 describes what this paper brings to the debate well. Point 1: I would question whether either psychology or sociology within the developmental social sciences in fact reflects gender-parity (L87).
We checked this in multiple ways.
First, we scanned departments that are ranked closely to Cornell in the US News and World Report ranking (setting aside our opinions of that ranking), and we counted the male/female ratio of Associate Professors or higher as a proxy for letter-writers (as nearly no writers were assistant professors). This agreed quite well with who did write the letters as shown by our rank plot in Figure 3.) These schools actually accounted for a significant fraction of the letters, and were consistent with the entire population from the NSF reports we referenced. Those are reported in Table 3 (for data, see lines 850-854). We further (not reported) tried to zero in on the subset of developmentally-oriented faculty, which was difficult because of the overlaps with other specializations and subject to interpretation, but obtained similar results. We followed the same procedure in EPP – we know the “feeder” universities for Fermilab. The web pages were easily accessible, and it was simple just to count.
The aggregate data also agree. The NSF 2017 study of doctoral recipients 19-301 Table 16 shows that 83.1% of developmentally-oriented PhDs went to women and 15.8% of PhDs in particle physics went to women. These numbers are not necessarily the same as the pool of writers, since these numbers refer to candidates, which is why we also estimated the fractions from scanning department web pages. The precise numbers are not critical to our argument, but the data support our claim.
Point 2 (LL90-104): correct approach but a side question: would either the gender of the primary advisors or their relative weight/reputation make a difference? And was there a difference between candidates applying within schools as opposed to external to schools? (may influence choice of referee writers) [in part this is covered in 1.1.2 point3.]
We addressed the rank (weight/reputation) issue in Fig 3b. We have looked at this in several other ways that we did not believe merited coverage in the paper. We did not examine whether writers at different ranks wrote more or less gender-biased letters, but since the F/M writer rank distributions are almost identical, we do not believe this can be a significant effect on our reported results. With respect to in-school vs. out-of-school: at Cornell, there were very few in-school applications. At Fermilab, there is a great deal of motion back and forth from Fermilab to individual universities, and out-of-Fermilab writers collaborate closely with Fermilab physicists, so any “in-out” distinction was reduced by familiarity. In any case, we cannot think of how this would affect our lexical results.
Results thoroughly described. One question that jumps out: was analysis done looking at differences in letter content of successful candidates and unsuccessful candidates?
Indeed this was the 800 lb. gorilla in the room from the beginning of this project, and we discussed various experimental approaches to address it, all requiring external grant funding. We can say more about this if requested, as we have a draft proposal to NSF to answer this very question, but it will require resources beyond our current ability.
For now, we addressed this in new text in the paper. Once you move to selection, many factors that have nothing to do with lexical content come into play. We did originally suggest the right way to answer this fascinating question was to write identical letters with swapped gendered names and assess how they were evaluated. Aside from this being an entirely new study outside the scope of the paper, there is another problem. EPP is so small that one could not effectively hide the applicant’s identity since their research and accomplishments would make it easy to determine who they were. For all these reasons, we decided not to cross the line to evaluation or even report easily misinterpreted results. Please see lines 777-801 of the revised paper for a fuller response.
The discussion on syntactic differences (2.3.2 para 1) and the effect on semantic analysis is a key topic that might reward further analysis at some stage. One observation related to lack of difference in EPP (lines 380-382) relates also to the point above: EPP writers are much more likely to be formulaic than DSS writers, which, if put alongside issues described in 3.4.2, would suggest the difference observed here would be expected.
The first author, who has experience chairing EPP hiring committees for many years, has not found the letters to be formulaic, and having read all of the social science letters, does not find the EPP letters to be more formulaic than the social science letters. Unfortunately, we cannot share the richness, nuance, stylistic differences, and passion expressed in the EPP letters.
Figure 2: is there a missing “of”? (four additional of eight)
fixed, thanks.
Section 3.5 is in many ways the most important section for three reasons: it identifies potential limitations in the analytical power of the techniques applied here, it identifies a major difference between fields and it points to an area of future research. It also suggestions a question, also suggested by the conclusion, that there may be differences in the relationship between the writer and expectations of the field of study/employment on the one hand and the relationship between the writer and the candidate on the other which may have a bearing on the context and outcomes.
We agree with the reviewer’s astute insight. It is the reason we recognized that we needed external resources to pursue it. This manuscript, although it has taken us several years to complete, is the first step and we agree with the reviewer that it is an important issue that deserves to be addressed later. This is why we focused strictly on the literature claiming that letters written for women were different (inferior) from those written for men. We have implemented new analytic techniques to explore these former claims, which will serve as the background for experimental studies later.
Line 588: is this supposed to be “scienc*” or, as on page 19 “scien*”?
fixed, thanks. It’s “scienc*”.
Does the paper achieve what it sets out to do as described in 1.1.2? The short answer is yes. Does it point to pathways for future research and remaining gaps? Yes.
In response to your other comments, we added a discussion of the age and number of post-docs of the candidates starting at line 180. EPP candidates usually had two post-docs, sometimes one, never zero, which differs from the typical (we wrote modal in the paper to be more precise) social science candidate who had zero postdocs; zero or one was the overwhelming majority. This is an interesting point that may affect how letters are written, and although the relation to gender bias is not clear, it is definitely worth reporting and we appreciate your bringing it up.
And thank you for a careful and useful review.

Reviewer 3 Report
The authors discuss gender bias in letters of recommendations for academic jobs in two disciplines. In doing so they address a specific problem which contributes to the underrepresentation of women in male dominated fields like physics. The paper is based on an adequate sample of recommendation letters from two disciplines – one with high women representation, one where women are underrepresented. The paper focuses on three questions related to gender bias and one related to methodological limitations of the approach: (1) Is there a gender bias in the selection of writers of recommendation letters? (2) Is there a gender bias in the wording? (3) Is there a bias in the rank of the writers? (4) Are there limitations in using pre-existing word lists?
The main strength of the paper is its methodological approach event though the analysis remains at a descriptive level. However, the hypotheses to be tested are not explicated in the paper and the extent to which the paper – and especially the discussion of the results – is embedded in the existing literature remains limited. The presentation of results would profit from explicating the hypotheses to be tested (e.g. at the end of the introduction). Some formulations contain implicit hypotheses – e.g. in section 3.1. (lines 327-328) “given the striking underrepresentation of women in EEP, we were surprised not to find weaker letters for women”.
Furthermore, the paper would profit from integrating the relevant results of the literature study in the introduction (hypotheses derived from the state of the art) and in the discussion of results. It is a bit surprising for readers, to have the main results of previous studies hidden in an annex.
When reading the paper, I wondered why authors did not integrate the variable “successful application” in the analysis. As far as I understood the sample represents letters of recommendations for applications in a specific institute. Probably it is known, which candidates have been selected. It would be interesting to know if there is a gender bias in letters of recommendations which lead to positive selection. This would probably allow to discuss the effect of gendered letters of recommendation on the selection process.
Some minor points which should be clarified to support readers:
- Table 1 – descriptive statistics for sample size: it is not clear for me what “unique male” and “unique female” refers to. This variable is not used in other tables and not specifically taken up in the interpretation.
- There is a deviation of EPP Total male/male writers between Table 1 and Table 4 – 842 and 844.
- It should be made transparent where assumptions come from – e.g. the assumption that 5% to 20% of reported gender differences could be false positive (line 300) or the assumption that letter-length is a proxy for enthusiasm (line 352)
It is an interesting paper, but it is not clear if it aims at contributing to the methodological discussion or to the discussion about gender bias in selection processes and the contribution of recommendation letters to this bias. There is large potential for both foci. However, depending on the focus the structure and the main arguments should be revised.
Author Response
Reviewer 3:
Furthermore, the paper would profit from integrating the relevant results of the literature study in the introduction (hypotheses derived from the state of the art) and in the discussion of results. It is a bit surprising for readers, to have the main results of previous studies hidden in an annex.
We have an extensive section on prior results. We do agree that our approach was to be descriptive rather than interpretive and deliberately avoided discussions of theory. We tried various ways of discussing earlier work and settled on our text; nothing is perfect.
When reading the paper, I wondered why authors did not integrate the variable “successful application” in the analysis. As far as I understood the sample represents letters of recommendations for applications in a specific institute. Probably it is known, which candidates have been selected. It would be interesting to know if there is a gender bias in letters of recommendations which lead to positive selection. This would probably allow to discuss the effect of gendered letters of recommendation on the selection process.
Indeed, as noted above, this was the 800lb gorilla in the room from the beginning of this project and we discussed various experimental approaches to address it, all requiring external grant funding. We can say more about this if requested, as we have a draft proposal to NSF to answer this very question, but it will require resources beyond our current ability.
For now, we addressed this in new text in the paper. Once you shift the research away from gender differences in lexical analysis of letters to actual hiring, many factors come into play that have nothing to do with lexical content. We did originally suggest the right way to answer this very interesting question was to write identical letters with swapped gendered names. Aside from this being an entirely new study outside the scope of the paper, there is another problem. EPP is so small that one could not effectively hide the identity of the applicant since their research and accomplishments would make it easy to determine who they were. For all these reasons we decided not to cross the line to evaluation or even report easily misinterpreted results. Please see lines 777-801 of the revised paper for a fuller response.
Some minor points which should be clarified to support readers:
- Table 1 – descriptive statistics for sample size: it is not clear for me what “unique male” and “unique female” refers to. This variable is not used in other tables and not specifically taken up in the interpretation.
Our fault. The EPP sample had about 10% of candidates that applied multiple times, and in these cases we removed all but the last application. “Unique” was supposed to refer to that. We removed the word and explained what we did in the caption.
- There is a deviation of EPP Total male/male writers between Table 1 and Table 4 – 842 and 844.
A typo, fixed. Thanks!
- It should be made transparent where assumptions come from – e.g. the assumption that 5% to 20% of reported gender differences could be false positive (line 300) or the assumption that letter-length is a proxy for enthusiasm (line 352)
This is a calculation from the number of variables and the p = 0.05 cut; 5% for one variable, 20% for four, since they are independent. We thought it not worth a more sophisticated calculation since our main point was that false positives were a potential issue. Next, we did not use the notion that letter-length is a proxy for enthusiasm, but simply mentioned it as a motivation – the results we report would not change. Others have already made this linkage: Dutt (2016) is just one paper that examined letter length as a proxy for enthusiasm and we changed the paper to point out extant literature has examined letter length, so we did as well.
Nevertheless, after reading more than 1000 letters in each of two fields, and having written hundreds of letters among ourselves, we know we write longer letters for candidates we want to be successful. The difference is apparent in very short, perfunctory letters. Ones that more or less recite the candidate’s accomplishments tend to be short; ones that then go into detailed praise are longer. But again, this notion is a motivation for looking at the variable, and our results do not change: women social scientists write longer letters than male social scientists.
It is an interesting paper, but it is not clear if it aims at contributing to the methodological discussion or to the discussion about gender bias in selection processes and the contribution of recommendation letters to this bias. There is large potential for both foci. However, depending on the focus the structure and the main arguments should be revised.
Here we agree that the paper has two foci, one on gender bias and one on methodology. We began by doing exactly (or as close as we could get to) what has been done in the past and then moved beyond it with new methods. We think the structure and main arguments presented as we wrote them are the best way to express our work.
And thank you for a careful and useful review.
Round 2
Reviewer 3 Report
The comments made in the first review have been considered which improved the paper. I do not have additional comments.